# Optical dopamine monitoring with dLight1 reveals mesolimbic phenotypes in a mouse model of neurofibromatosis type 1

J Elliott Robinson[1], Gerard M Coughlin[1], Acacia M Hori[1], Jounhong Ryan Cho[1], Elisha D Mackey[1], Zeynep Turan[1], Tommaso Patriarchi[2], Lin Tian[2], Viviana Gradinaru[1]*

[1]Division of Biology and Biological Engineering, California Institute of Technology, Pasadena, United States; [2]Department of Biochemistry and Molecular Medicine, University of California, Davis, Davis, United States

**Abstract** Neurofibromatosis type 1 (NF1) is an autosomal dominant disorder whose neurodevelopmental symptoms include impaired executive function, attention, and spatial learning and could be due to perturbed mesolimbic dopaminergic circuitry. However, these circuits have never been directly assayed in vivo. We employed the genetically encoded optical dopamine sensor dLight1 to monitor dopaminergic neurotransmission in the ventral striatum of NF1 mice during motivated behavior. Additionally, we developed novel systemic AAV vectors to facilitate morphological reconstruction of dopaminergic populations in cleared tissue. We found that NF1 mice exhibit reduced spontaneous dopaminergic neurotransmission that was associated with excitation/inhibition imbalance in the ventral tegmental area and abnormal neuronal morphology. NF1 mice also had more robust dopaminergic and behavioral responses to salient visual stimuli, which were independent of learning, and rescued by optogenetic inhibition of non-dopaminergic neurons in the VTA. Overall, these studies provide a first in vivo characterization of dopaminergic circuit function in the context of NF1 and reveal novel pathophysiological mechanisms.

*For correspondence:
viviana@caltech.edu

Competing interests: The authors declare that no competing interests exist.

## Introduction

Neurofibromatosis type 1 (NF1) is an autosomal dominant disorder of neural crest-derived tissues that affects approximately 1 in 3500 individuals worldwide and is caused by loss of one functional copy of the *NF1* gene on chromosome 17 (*Wallace et al., 1990*). Neurofibromin, the protein product of *NF1*, inhibits Ras-dependent cellular growth and proliferation (*Basu et al., 1992*) and enhances cAMP signaling pathways (*Tong et al., 2002*). The clinical features of NF1 include pigmentary lesions, neoplasia (e.g. cutaneous and plexiform neurofibromas, optic gliomas, malignant peripheral nerve sheath tumors), cognitive and learning disabilities, peripheral neuropathy, musculoskeletal abnormalities, and gross and fine motor delays (*Cimino and Gutmann, 2018*; *Gutmann et al., 2012*). Cognitive dysfunction is a significant source of lifetime morbidity, as up to 70% of affected individuals experience impaired executive functioning, speech and language delays, attention deficits, hyperactivity, and/or impulsivity (*Hyman et al., 2005*). Furthermore, approximately one third of patients with NF1 meet DSM-V criteria for attention deficit hyperactivity disorder (ADHD) (*Hyman et al., 2005*; *Miguel et al., 2015*). Despite the societal burden of NF1-associated cognitive sequelae, their etiology has not been fully elucidated.

Although homozygous genetic disruption of the *Nf1* gene is embryonic lethal in mice (*Silva et al., 1997*), cognitive deficits in NF1 have been successfully modeled in several transgenic and

**eLife digest** About one in 3,500 people have a genetic disorder called neurofibromatosis type 1, often shortened to NF1, making it one of the most common inherited diseases. People with NF1 may have benign and cancerous tumors throughout the body, learning disabilities, developmental delays, curvature of the spine and bone abnormalities. Children with NF1 often experience difficulties with attention, hyperactivity, speech and language delays and impulsivity. They may also have autism spectrum disorder, or display symptoms associated with this condition.

Studies in mice with a genetic mutation that mimics NF1 suggest that abnormal development in cells in the middle of the brain may cause the cognitive symptoms. These midbrain neurons produce a chemical called dopamine and send it throughout the brain. Dopamine is essential for concentration and it is involved in how the brain processes pleasurable experiences.

Now, Robinson et al. show that, at rest, the NF1 model mice release dopamine less often than typical mice. This happens because, when there are no stimuli to respond to, neighboring cells slow down the activity of dopamine-producing neurons in NF1 model mice.

In the experiments, both NF1 model mice and typical mice were taught to associate environmental cues with rewards or punishments. Robinson et al. then measured the release of dopamine in the mice using a sensor called dLight1, which produces different intensities of fluorescent light depending on the amount of dopamine present. This revealed that the NF1 model mice produced more dopamine in response to visual cues and had enhanced behavioral responses to these stimuli. For example, when a looming disc that mimics predators approached them from above, the NF1 model mice tried to hide in an exaggerated way compared to the typical mice. Previously, it had been shown that this type of behavior is due to the activity of the dopamine-producing neurons' neighboring cells, which Robinson et al. found is greater in NF1 model mice.

Next, Robinson et al. stopped neighboring cells from interfering with the dopamine-producing neurons in NF1 model mice. This restored dopamine release to normal levels at rest, and stopped the mice from overreacting to the looming disc. The experiments help explain how the NF1 model mice process visual information. Further study of the role dopamine plays in cognitive symptoms in people with NF1 may help scientists develop treatments for the condition.

conditional knockout mouse lines (*Silva et al., 1997*; *Zhu et al., 2001*; *Hegedus et al., 2007*; *Cui et al., 2008*; *Brown et al., 2010a*; *Anastasaki et al., 2015*; *Omrani et al., 2015*; *Li et al., 2016*; *Xie et al., 2016*). Heterozygous knockout mice ($Nf1^{+/-}$) exhibit impaired spatial learning (*Costa et al., 2001*; *Silva et al., 1997*), which is Ras/ERK-dependent (*Costa et al., 2002*), rescued by the Ras inhibitor lovastatin (*Li et al., 2005*), and may be due to increased inhibitory GABA tone (*Costa et al., 2002*). Additionally, the neurofibromin C-terminus is a positive regulator of G-protein-stimulated adenylyl cyclase activity (*Hannan et al., 2006*; *Tong et al., 2002*), and cAMP deficiency in NF1 knockout models causes altered in vitro neuronal morphology and growth, visual learning deficits, and changes in cortical architecture in mice (*Brown et al., 2012*; *Brown et al., 2010b*; *Hegedus et al., 2007*; *Wolman et al., 2014*). Attenuated dopaminergic neurotransmission in meso-limbic and nigrostriatal circuits are putative mechanisms underlying attentional, learning, and motivational deficits observed in NF1 model mice (*Diggs-Andrews and Gutmann, 2013*). Mesolimbic reward circuits involve the convergence of dopaminergic projections from the midbrain ventral tegmental area (VTA) with glutamatergic inputs from cortical and subcortical regions on medium spiny neurons in the nucleus accumbens (NAc). These circuits facilitate the translation of relevant internal and external stimuli into motivated behaviors (*Wise, 2005*) and have been implicated in the pathophysiology of ADHD and other disorders of impulse control (*Li et al., 2006*; *Purper-Ouakil et al., 2011*).

In the optic glioma mouse model of NF1 (OPG, a conditional *Nf1* knockout in astrocytes on an $Nf1^{+/-}$ background), reduced striatal dopamine is associated with motor, exploratory, spatial learning, and attentional abnormalities (*Brown et al., 2010a*; *Diggs-Andrews et al., 2013*; *Anastasaki et al., 2015*), which are ameliorated by treatment with the catecholamine re-uptake inhibitor methylphenidate or the dopamine precursor L-DOPA (*Brown et al., 2010a*). Despite these efforts, dopaminergic neurotransmission has never been investigated in NF1 models in vivo. In order

to address this gap in the understanding of NF1, we utilized the new, ultra-fast, genetically encoded dopamine sensor dLight1 (*Patriarchi et al., 2018*) to monitor dopamine dynamics in the lateral nucleus accumbens (LNAc) during motivated behavior in 129T2/SvEmsJ::C57Bl/6NTac F1 hybrid *Nf1* wildtype ($Nf1^{+/+}$) and heterozygous knockout ($Nf1^{+/-}$) mice. This hybrid background produces more robust behavioral phenotypes than those on a pure C57Bl/6 background (*Cui et al., 2008*; *Li et al., 2005*; *Shilyansky et al., 2010*). Novel dopaminergic phenotypes were further parsed with patch clamp electrophysiology and optogenetics. Because previous morphological analysis has largely been restricted to neuronal cultures (*Brown et al., 2010a*; *Anastasaki et al., 2015*), we comprehensively characterized dopaminergic neuron structure in situ in $Nf1^{+/+}$ and $Nf1^{+/-}$ mice using tissue clearing, tracing methods, and the novel systemic AAV-based tool *Th*-VAST (catecholaminergic neuron-targeted vector-assisted spectral tracing). These efforts revealed distinct dopaminergic phenotypes, identified putative mechanisms governing their expression, and explored how *Nf1* haploinsufficiency moderates the motivational salience of relevant environment stimuli.

## Results

### In vivo optical monitoring of dopaminergic neurotransmission using dLight1.2

In order to investigate dopamine dynamics in freely behaving $Nf1^{+/+}$ and $Nf1^{+/-}$ mice, we utilized the genetically encoded, fluorescent dopamine sensor dLight1.2 (*Patriarchi et al., 2018*), which allows for sub-micromolar detection of extracellular dopamine concentrations with sub-second resolution and negligible sensitivity to other monoamines, GABA, and glutamate (*Corre et al., 2018*; *Patriarchi et al., 2018*). Fluorescent dopamine signals in the LNAc were monitored with fiber photometry (*Gunaydin et al., 2014*); this terminal field region was chosen because its afferent ventral tegmental dopaminergic inputs exhibit a high diversity of responses to both rewarding and aversive stimuli and stimulus-predictive cues (*de Jong et al., 2019*; *Lammel et al., 2011*). To facilitate optical dopamine measurements, an adeno-associated viral vector (AAV9-hSyn-dLight1.2) was stereotaxically injected into the LNAc to express dLight1.2 in neurons, followed by implantation of a 400 μm optical fiber (*Figure 1A*) for sensor excitation and emitted photon collection via a custom photometry system (*Cho et al., 2017*) (*Figure 1B*).

After surgical recovery, we measured baseline differences in spontaneous dopaminergic neurotransmission by monitoring dLight1.2 signals (*Figure 1C*, *Figure 1—figure supplement 1*) in the LNAc during 5-min epochs in which mice sat in a dark, sound-attenuating chamber. Peak analysis was performed to identify local trace prominences (*Figure 1D*) and revealed that the dopamine transient event rate was reduced in $Nf1^{+/-}$ mice compared to $Nf1^{+/+}$ littermates (*Figure 1E*). Baseline (median) fluorescence, peak amplitude, and full width at half maximal intensity (FWHM) was equivalent between genotypes. Because reduced LNAc dopamine content and afferent terminal TH expression have been observed in OPG mice (*Brown et al., 2010a*; *Diggs-Andrews et al., 2013*), we measured monoamine and monoamine metabolite levels in the NAc using high-performance liquid chromatography. We failed to detect differences in dopamine (DA), serotonin (5-HT), norepinephrine (NE), or their metabolites between genotypes (*Figure 1—figure supplement 2*). Additionally, there was no difference in dopaminergic terminal tyrosine hydroxylase expression across striatal sub-compartments (*Figure 1—figure supplement 2*). These findings suggest that basal differences in dLight1.2 event rate are not due to changes in dopaminergic terminal density or dopamine synthetic capacity.

In order to further parse differences in spontaneous dopaminergic transient activity, we performed whole-cell patch clamp electrophysiological recordings in acute midbrain slices that contained the lateral ventral tegmental area (*Figure 1F*), which is the main source of dopaminergic projections to the LNAc (*Lammel et al., 2011*). Because the dependence of $Nf1^{+/-}$ phenotypes on genetic background precludes crossing with cell-type-specific reporter or Cre recombinase lines, we used a blood-brain barrier penetrant, systemic adeno-associated viral vector (AAV-PHP.eB) (*Chan et al., 2017*) containing a green fluorescent protein (GFP) transgene under control of the rat tyrosine hydroxylase promoter (*Oh et al., 2009*) (AAV-PHP.eB-*Th*-GFP; $1 \times 10^{11}$ viral genomes/mouse r.o.; *Figure 1F*, *right*) to label dopaminergic neurons. This allowed for visual identification during patch clamp experiments. GFP-positive cells were considered to be dopaminergic if their

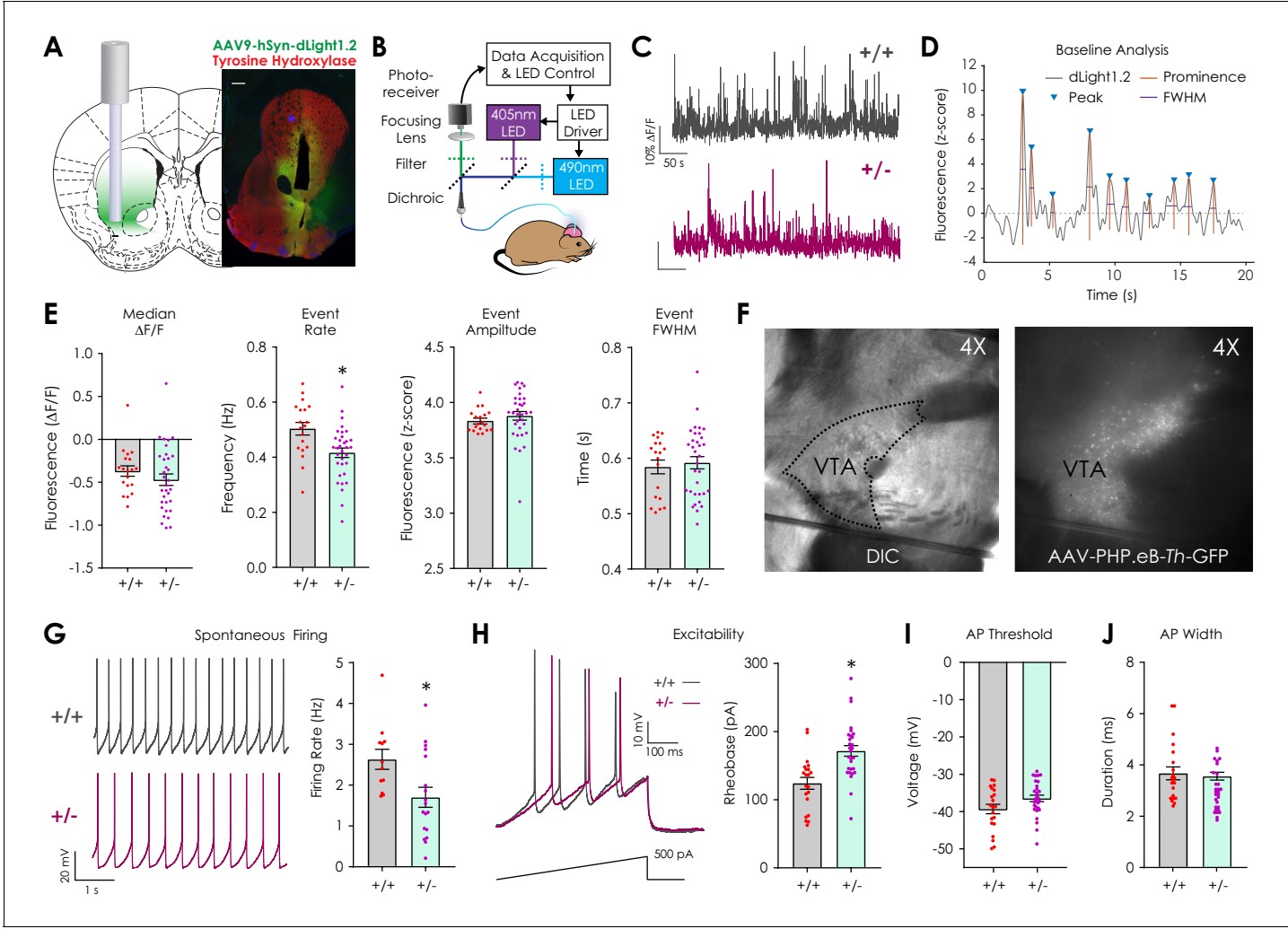

**Figure 1.** Assessment of basal dopaminergic function in vivo with dLight1.2 and ex vivo patch clamp electrophysiology. (A) Illustration showing location of stereotaxic injection of the AAV9-hSyn-dLight1.2 viral vector and photometry fiber implantation (*left*). Representative histological image (*right*, scale: 300 μm) showing the fiber tip location and expression of dLight1.2 (stained for GFP, green) and dopaminergic terminal tyrosine hydroxylase (TH, Red). (B) Schematic of fiber photometry system used for dLight1.2 (490 nm) and isosbestic (405 nm; reference signal) excitation and emission signal detection in freely moving mice. (C) Representative dLight1.2 traces in $Nf1^{+/+}$ and $Nf1^{+/-}$ mice. (D) Representative trace and analysis features for baseline peak detection. (E) Peak analysis of baseline dLight1.2 recordings revealed that Nf1$^{+/-}$ mice (n = 33) exhibit reduced transient frequency (unpaired t-test; $t_{50} = 3.06$, p=0.004) but not median fluorescence (unpaired t-test; $t_{50} = 1.01$, p=0.32), transient amplitude (unpaired t-test; $t_{50} = 0.83$, p=0.41), or full width at half maximal amplitude (FWHM; unpaired t-test; $t_{50} = 0.43$, p=0.67) when compared to Nf1$^{+/+}$ littermates (n = 19). (F) 4X differential interference contrast (DIC) image (*left*) of an acute horizontal midbrain slice containing the ventral tegmental area (VTA) and 4X epifluorescence image (*right*) with GFP-labeled catecholaminergic neurons following systemic delivery of AAV-PHP.eB-*Th*-GFP (1 × 10$^{11}$ v.g./mouse). (G) Representative traces showing spontaneous whole-cell firing of putative VTA dopaminergic neurons (*left*). Spontaneous firing rates (*right*) were lower (unpaired t-test; $t_{28} = 2.58$, p=0.0 w) in $Nf1^{+/-}$ putative dopaminergic neurons (n = 18) compared to $Nf1^{+/+}$ neurons (n = 12). (H) Representative electrophysiological traces (*left*) showing evoked firing by a 1 pA/ms ramp current from −60 mV in $Nf1^{+/+}$ and $Nf1^{+/-}$ putative dopaminergic neurons. Rheobase (*right*; unpaired t-test; $t_{48} = 4.05$, p<0.001) but not action potential threshold (I; $t_{48} = 1.93$, p=0.06) or width (J; $t_{48} = 0.39$, p=0.70) was increased in $Nf1^{+/-}$ (n = 29) putative dopaminergic neurons compared to $Nf1^{+/+}$ (n = 21). *denotes p<0.05 vs $Nf1^{+/+}$. Data presented as mean ± SEM.

The online version of this article includes the following figure supplement(s) for figure 1:

**Figure supplement 1.** Raw fluorescent photometry signals.

**Figure supplement 2.** Striatal catecholamine content and tyrosine hydroxylase immunofluorescence.

action potential duration was >1 ms, a previously validated threshold to distinguish dopaminergic from GABAergic neurons in the VTA (*Chieng et al., 2011*). We found that putative dopaminergic neurons in $Nf1^{+/-}$ midbrain slices exhibited lower spontaneous whole-cell firing rates (*Figure 1G*) and required more rheobase current to elicit a spike when compared to $Nf1^{+/+}$ neurons (*Figure 1H*).

This finding supports the hypothesis that phenotypic differences in baseline dLight1.2 event metrics are activity-dependent. Action potential threshold, duration, amplitude, and after hyperpolarization magnitude did not differ between genotypes (*Figure 1I–J*, *Table 1*).

## Morphological characterization of VTA dopaminergic neurons in *Nf1*$^{+/+}$ and *Nf1*$^{+/-}$ mice

During whole-cell recordings, we also observed that *Nf1*$^{+/-}$ putative dopaminergic neurons exhibit increased input resistance ($R_m$) and decreased membrane capacitance ($C_m$) compared to *Nf1*$^{+/+}$ littermates (*Figure 2A*) without a change in other membrane properties (*Table 2*). This finding was robust across experiments (*Table 2*). Because increased $R_m$ could be indicative of reduced soma volume (*Torres-Torrelo et al., 2014*), we manually traced over two thousand TH-positive dopaminergic somata in the VTA per genotype (*Figure 2B*). We found that cross-sectional area, major axis length, and minor axis length were reduced in *Nf1*$^{+/-}$ mice (*Figure 2C–D*, *Figure 2—figure supplement 1*). Proportionality was maintained, however, as the soma aspect ratio was equivalent between genotypes (*Figure 2—figure supplement 1*). TH immunofluorescence and total neuron counts in the VTA did not differ between *Nf1*$^{+/-}$ and *Nf1*$^{+/+}$ dopaminergic neurons (*Figure 2D*). No phenotypic differences were observed in the adjacent substantia nigra pars compacta (*Figure 2—figure supplement 1*). These findings indicate that relative differences in soma size were VTA-specific and could have contributed to changes in passive membrane properties.

Dendritic complexity also contributes to cell input resistance (*Bekkers and Hausser, 2007*; *Šišková et al., 2014*), so we modified the two-component, systemic AAV-based method VAST (Vector-Assisted Spectral Tracing) (*Chan et al., 2017*) to create *Th*-VAST. This tool facilitates anatomical reconstruction of dendritic arbors by providing recombinase-independent, sparse, multicolor labeling of catecholaminergic neurons. VAST achieves hue diversity via stochastic expression of three tetracycline response element (TRE)-regulated fluorescent proteins (XFPs; mRuby2, mNeonGreen, and mTurquoise2) following systemic delivery with AAV-PHP.eB. Sparseness is subsequently tuned by titration of a co-delivered, tet-off transactivator (tTA) inducer vector (*Chan et al., 2017*). In *Th*-VAST, tTA expression is targeted to catecholaminergic neurons via use of the *Th* promoter, and retro-orbital delivery of the XFP cocktail (AAV-PHP.eB-TRE-XFP; $1 \times 10^{12}$ vg/mouse total) and the inducer vector (AAV-PHP.eB-*Th*-tTA; $1 \times 10^{11}$ vg/mouse) produced dense multicolor labeling of *Th* neurons in the VTA and SNc (*Figure 2E–F*, *Figure 2—figure supplement 2*). Compared to $1 \times 10^{12}$ vg/ mouse AAV-PHP.eB-*Th*-GFP (*Chan et al., 2017*), the specificity of *Th*-VAST vectors was lower in the VTA (58.7% vs 81%) and SNc (74.2% vs 81%) (*Figure 2—figure supplement 2*) despite good XFP restriction to these areas. This likely occurred because induction of XFP expression requires very low levels of tTA, and a sub-population of VTA projection neurons have hybrid *Th*-GABAergic phenotypes (*Root et al., 2014*; *Stuber et al., 2015*). As such, spectral tracing was only performed when *Th*-VAST-labeled neurons were unequivocally tyrosine hydroxylase-positive.

**Table 1.** Action potential features across patch clamp electrophysiology experiments.

| Property | Experiment | p | +/+: Mean ± SEM, n | +/-: Mean ± SEM, n |
|---|---|---|---|---|
| Rheobase | Baseline characterization | <0.001 | 124.1 ± 8.65 pA, n = 21 | 171.7 ± 7.779 pA, n = 29 |
| AP Threshold | Baseline characterization | 0.059 | −39.32 ± 1.266 mV, n = 21 | −36.45 ± 0.8708 mV, n = 29 |
| AP Duration | Baseline characterization | 0.695 | 3.671 ± 0.2525 ms, n = 21 | 3.562 ± 0.1499 ms, n = 29 |
| AP Height | Baseline characterization | 0.555 | 60.89 ± 1.607 mV, n = 21 | 59.42 ± 1.749 mV, n = 29 |
| AP AHP | Baseline characterization | 0.897 | −15.43 ± 1.19 mV, n = 21 | −14.88 ± 1.046 mV, n = 29 |
| Firing Rate | Baseline characterization | 0.016 | 2.633 ± 0.2464 Hz, n = 12 | 1.703 ± 0.244 Hz, n = 18 |
| Rheobase | Picrotoxin rescue | <0.001 | 131.5 ± 7.537 pA, n = 25 | 89.14 ± 6.413 pA, n = 20 |
| AP Threshold | Picrotoxin rescue | 0.456 | −36.92 ± 1.193 mV, n = 25 | −38.33 ± 1.472 mV, n = 20 |
| AP Duration | Picrotoxin rescue | 0.610 | 4.156 ± 0.1589 ms, n = 25 | 4.03 ± 0.1891 ms, n = 20 |
| AP Height | Picrotoxin rescue | 0.946 | 56.16 ± 2.021 mV, n = 25 | 55.99 ± 1.151 mV, n = 20 |
| AP AHP | Picrotoxin rescue | 0.168 | −13.84 ± 1.125 mV, n = 25 | −11.78 ± 0.844 mV, n = 20 |
| Firing Rate | Picrotoxin rescue | 0.714 | 2.434 ± 0.208 Hz, n = 16 | 2.535 ± 0.1596 Hz, n = 13 |

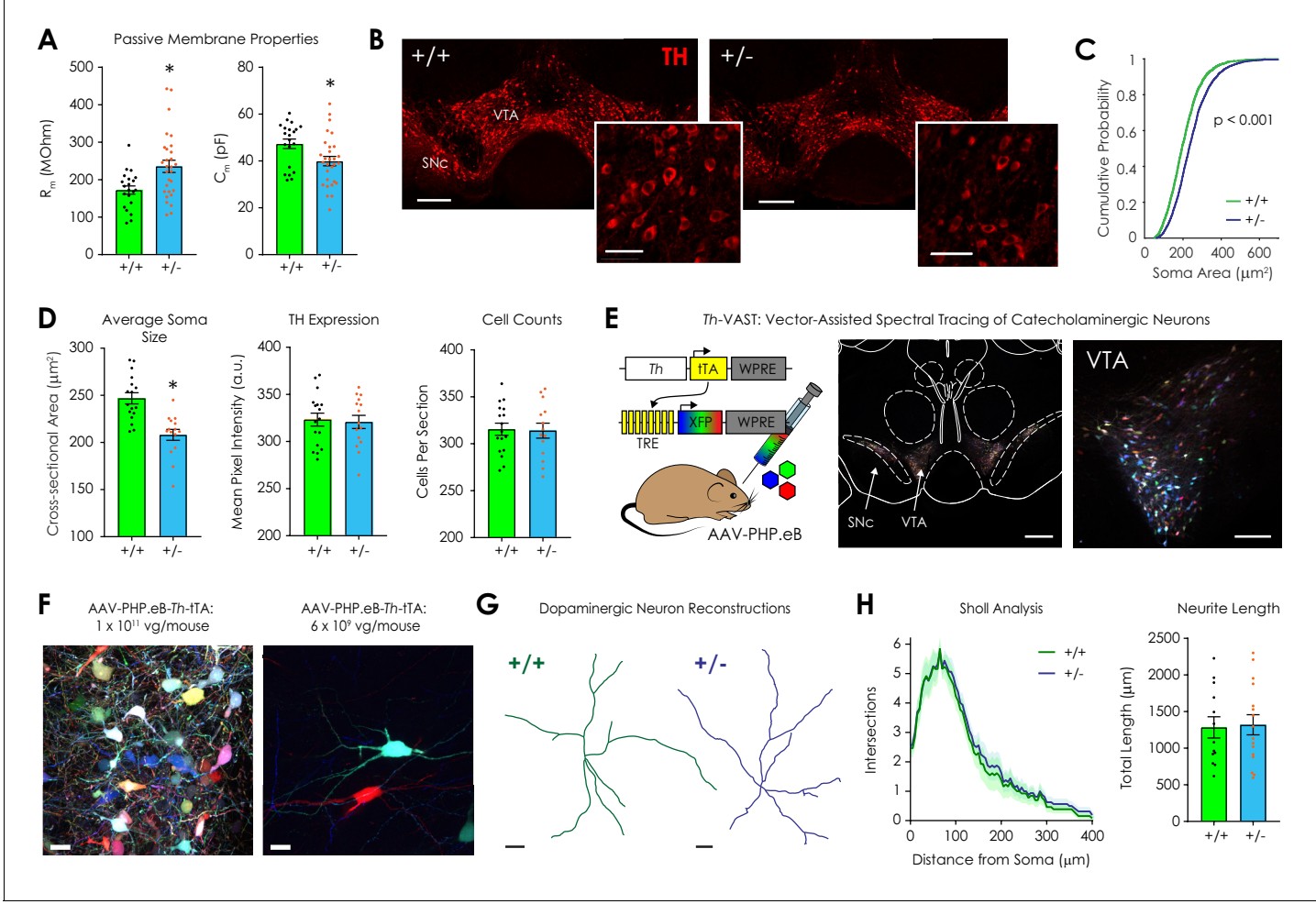

**Figure 2.** Morphological analysis of ventral tegmental dopaminergic neurons in $Nf1^{+/+}$ and $Nf1^{+/-}$ mice. (**A**) Whole-cell recordings revealed that $Nf1^{+/-}$ putative dopaminergic neurons (n = 29) had increased input resistance ($R_m$; *left*; unpaired t-test; $t_{48}$ = 2.97, p=0.005) and decreased capacitance ($C_m$; *right*; $t_{48}$ = 2.54, p=0.01) compared to $Nf1^{+/+}$ neurons (n = 21). (**B**) Representative ventral midbrain images containing the ventral tegmental area (VTA) and substantia nigra pars compacta (SNc) stained for tyrosine hydroxylase (TH, scale: 300 µm); TH-positive neurons in the VTA (*inset*, scale: 100 µm). (**C**) The cumulative probability distribution of the cross sectional area of manually traced $Nf1^{+/+}$ (n = 2344) and $Nf1^{+/-}$ (n = 2586) VTA dopaminergic neuron somata (two-sample Kolmogorov-Smirnov test; D = 0.18, p<0.001). (**D**) Average VTA dopaminergic soma area (*left*; $n_{+/+}$ = 17, $n_{+/-}$ = 15; unpaired t-test; $t_{30}$ = 4.65, p<0.001), TH immunofluorescence (*middle*; $t_{30}$ = 0.25, p=0.90), and number of neurons/histological section (*right*; $t_{30}$ = 0.15, p=0.88) per mouse. (**E**) *Th*-VAST (*left*) produced multicolor labeling of dopaminergic neurons in the VTA (*middle*, scale: 300 µm; *right*, scale: 100 µm). (**F**) Dense (*left*, scale: 20 µm) or sparse multi-color labeling (*right*, scale: 20 µm) was achieved via retro-orbital injection of either $1 \times 10^{11}$ or $6 \times 10^{9}$ vg/mouse AAV-PHP.eB-*Th*-tTA, respectively, and $1 \times 10^{12}$ total vg/mouse of the XFP cocktail (AAV-PHP.eB-TREx7-mRuby2, -mNeonGreen, or -mTurquoise2). (**G**) Representative dopaminergic neuron reconstructions following neurite tracing (scale: 20 µm). (**H**) Sholl analysis failed to detect a difference in dendritic complexity (*left*; two-way repeated measures ANOVA; $F_{80,2160}$ = 0.052, $p_{distance \times genotype}$ >0.99; $F_{80,2160}$ = 63.9, $p_{distance}$ <0.001; $F_{1,27}$ = 0.25, $p_{genotype}$ = 0.63) or total neurite length (*right*; unpaired t-test; $t_{27}$ = 0.18, p=0.86) between genotypes ($n_{+/+}$ = 13, $n_{+/-}$ = 16 for +/- group). * denotes p<0.05 vs $Nf1^{+/+}$. Data presented as mean ± SEM.

The online version of this article includes the following figure supplement(s) for figure 2:

**Figure supplement 1.** Additional data: histological analysis.
**Figure supplement 2.** Additional data: *Th*-VAST.

Using a lower inducer vector dose ($6 \times 10^{9}$ vg/mouse) to provide sparse labeling (**Figure 2F**, *right*), we repeated *Th*-VAST in $Nf1^{+/-}$ and $Nf1^{+/+}$ mice. Following two weeks of expression, we prepared and optically cleared (using RIMS) (**Yang et al., 2014**) 300 µm horizontal VTA sections that had been immunostained for TH to confirm post hoc that *Th*-VAST-labeled neurons were dopaminergic. After tracing in Imaris (**Figure 2G**), Sholl analysis was performed to quantify dendritic branching by detecting neurite intersections with concentric 5 µm shells originating from the soma. No

**Table 2.** Passive membrane properties across patch clamp electrophysiology experiments.

| Property | Experiment | p | +/+: Mean ± SEM, n | +/-: Mean ± SEM, n |
|---|---|---|---|---|
| $C_m$ | Baseline characterization | 0.014 | 47.35 ± 2.032 pF, n = 21 | 41.27 ± 2.026 pF, n = 29 |
| $R_m$ | Baseline characterization | 0.005 | 172.4 ± 10.94 MΩ, n = 21 | 235.7 ± 16.32 MΩ, n = 29 |
| $R_s$ | Baseline characterization | 0.966 | 17.86 ± 1.73 pF MΩ, n = 21 | 17.95 ± 1.257 MΩ, n = 29 |
| Holding | Baseline characterization | 0.658 | −74.26 ± 10.95 pA, n = 21 | −81.01 ± 10.18 pA, n = 29 |
| $C_m$ | $I_h$ measurement | 0.047 | 51.94 ± 4.45 pF, n = 14 | 42.53 ± 2.351 pF, n = 24 |
| $R_m$ | $I_h$ measurement | 0.009 | 170.8 ± 11.96 MΩ, n = 14 | 222.4 ± 12.49 MΩ, n = 24 |
| $R_s$ | $I_h$ measurement | 0.528 | 17.78 ± 1.478 MΩ, n = 14 | 19.1 ± 1.334 MΩ, n = 24 |
| Holding | $I_h$ measurement | 0.457 | −61.15 ± 8.657 pA, n = 14 | −52.11 ± 7.642 pA, n = 24 |
| $C_m$ | Picrotoxin rescue | 0.004 | 47.74 ± 2.276 pF, n = 29 | 36.62 ± 2.956 pF, n = 20 |
| $R_m$ | Picrotoxin rescue | 0.001 | 181 ± 8.464 MΩ, n = 29 | 239 ± 14.94 MΩ, n = 20 |
| $R_s$ | Picrotoxin rescue | 0.670 | 17.23 ± 1.054 MΩ, n = 29 | 18.04 ± 1.648 MΩ, n = 20 |
| Holding | Picrotoxin rescue | 0.611 | −56.77 ± 5.88 pA, n = 29 | −61.64 ± 7.639 pA, n = 20 |

difference in dendritic complexity or total neurite length was observed between genotypes (*Figure 2H*), which suggests that, although $Nf1^{+/-}$ dopaminergic neurons have smaller somata than $Nf1^{+/+}$ neurons, they have similar neurite morphology.

## $Nf1^{+/-}$ putative dopaminergic neurons exhibit excitation/inhibition imbalance in the ventral tegmental area

The observation that $Nf1^{+/-}$ dopaminergic neurons have reduced cross-sectional areas but higher rheobase requirement was unexpected, given that smaller neurons tend to be more excitable (*Torres-Torrelo et al., 2014*). In order to parse these differences, we first assayed dopaminergic $I_h$ currents, which contribute to the stability of spontaneous firing rates (*Neuhoff et al., 2002*; *Seutin et al., 2001*) and are attenuated in hippocampal interneurons in NF1 model mice (*Omrani et al., 2015*). $I_h$ was determined by quantifying the sag current produced by a series of hyperpolarizing voltage steps from −60 mV to −130 mV in voltage clamp (*Figure 3A*). We found that $Nf1^{+/-}$ dopaminergic neurons had smaller $I_h$ current amplitudes (*Figure 3A*, *Figure 3—figure supplement 1*) without a change in voltage dependence (*Figure 3B*; determined by tail current analysis) relative to $Nf1^{+/+}$ littermates. Differences in $I_h$ current amplitudes were not significant when normalized to the cell capacitance to account for cell size (*Figure 3C*, *Figure 3—figure supplement 1*), and maximum $I_h$ current amplitude was significantly correlated with $C_m$ across all animals and within genotypes (*Figure 3D*). Since reduced cAMP production, which has been associated with $Nf1^{+/-}$ neuronal phenotypes in vitro (*Brown et al., 2012*), could attenuate the $I_h$ current, we repeated $I_h$ measurements in $Nf1^{+/-}$ slices in the presence of the adenylyl cyclase activator forskolin. Addition of 20 µM forskolin to the bath solution did not significantly affect $I_h$ magnitude or voltage dependence in $Nf1^{+/-}$ putative dopaminergic neurons (*Figure 3—figure supplement 1*). Thus, changes in $I_h$ magnitude are likely cAMP-independent, reflective of smaller cell size, and unlikely to be the etiologic cause of reduced cell excitability.

Excitation/inhibition imbalance due to increased GABAergic tone is a hypothesized mechanism governing NF1-associated cognitive deficits (*Diggs-Andrews and Gutmann, 2013*), and $GABA_A$ receptor agonists increase rheobase (*Rojas et al., 2011*), so we investigated excitation/inhibition balance in putative dopaminergic neurons by measuring spontaneous inhibitory (sIPSC) and excitatory post-synaptic currents (sEPSC) in voltage clamp (*Figure 3E*). We found that $Nf1^{+/-}$ putative dopaminergic neurons displayed increased sIPSC frequency but not sIPSC amplitude, sEPSC frequency, or sEPSC amplitude compared to $Nf1^{+/+}$ neurons (*Figure 3F–G*). Addition of 100 µM picrotoxin (a highly selective, non-competitive $GABA_A$ receptor antagonist) to the bath solution rescued spontaneous firing rates in $Nf1^{+/-}$ putative dopaminergic neurons (*Figure 3H*) without affecting passive membrane properties (*Table 2*). Picrotoxin also reduced rheobase to levels significantly lower than control- and picrotoxin-treated $Nf1^{+/+}$ cells (*Figure 3I*), which would be expected at baseline due to differences in soma volume.

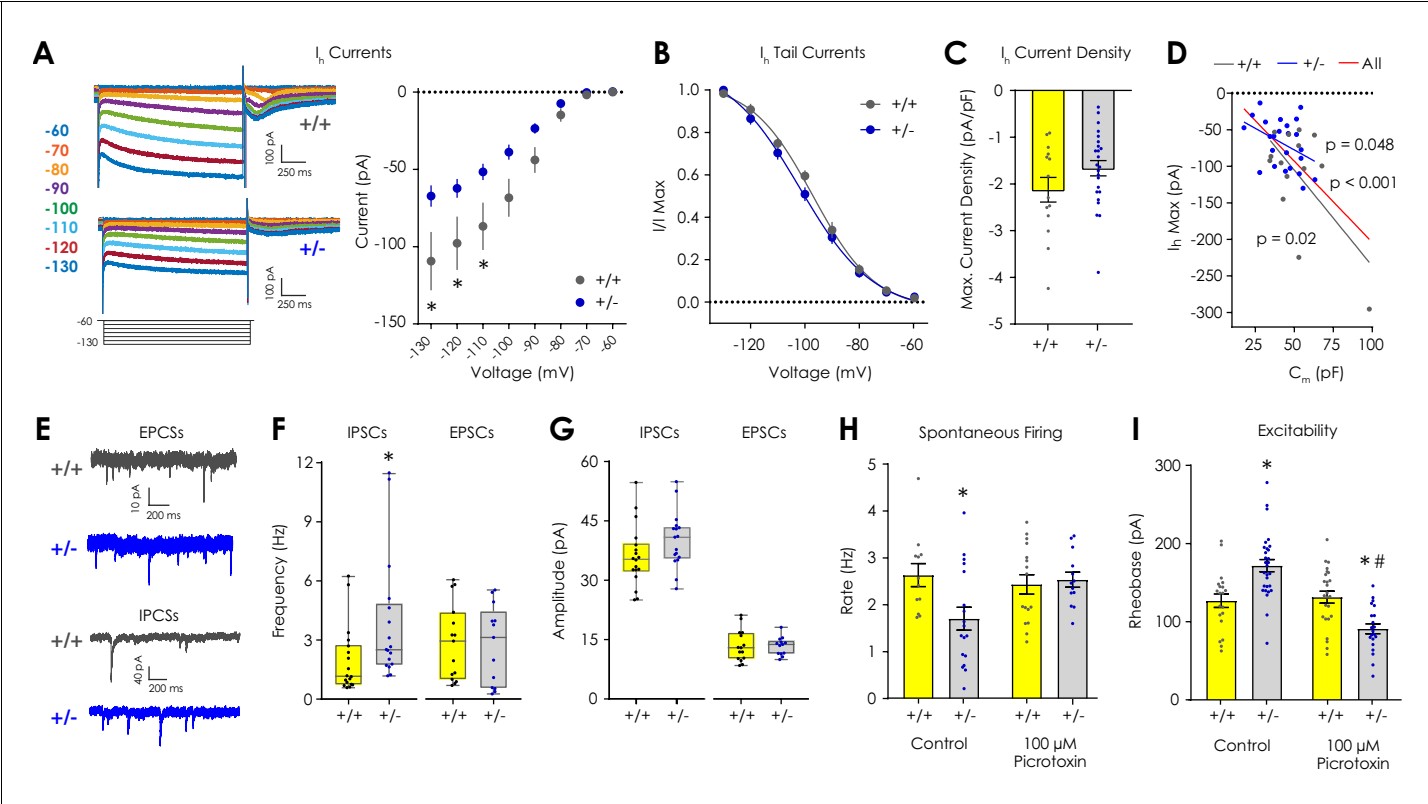

**Figure 3.** Electrophysiological characterization of $I_h$, inhibitory, and excitatory currents in VTA dopaminergic neurons ex vivo. (A) Representative traces showing $I_h$ currents during hyperpolarizing voltage steps from −60 to −130 mV. (B) $I_h$ current magnitude was smaller (2-way repeated measures ANOVA with Bonferroni post hoc tests; $F_{7,252}$ = 5.38, $p_{genotype \times voltage}$ <0.001) in $Nf1^{+/-}$ putative dopaminergic neurons (n = 24) compared to $Nf1^{+/+}$ neurons (n = 14). (B) Tail current analysis showed no difference in the $I_h$ voltage dependence between $Nf1^{+/+}$ (n = 14, EV50 = −96.98 mV, 95% CI = −99.69 to −94.52 mV) and $Nf1^{+/-}$ putative dopaminergic neurons (n = 24, EV50 = −101.9 mV, 95% CI = −106.5 to −98.86 mV). (C) Maximum $I_h$ current density did not differ between $Nf1^{+/+}$ (n = 14) and $Nf1^{+/-}$ (n = 24) putative dopaminergic neurons (unpaired t-test; $t_{36}$ = 1.56, p=0.13). (D) $I_h$ magnitude was negatively correlated with $C_m$ in $Nf1^{+/+}$ ($R^2$ = 0.39, p=0.02), $Nf1^{+/-}$ ($R^2$ = 0.17, p=0.049), and across all putative dopaminergic neurons ($R^2$ = 0.35, p<0.001). (E) Representative traces of spontaneous excitatory (sEPSC) and inhibitory (sIPSC) post-synaptic currents. (F) The frequency of sIPSCs ($n_{+/+}$ = 18, $n_{+/-}$ = 17; Mann-Whitney U test; $U$ = 74.5, p=0.009; unpaired t-test; $t_{33}$ = 2.20, p=0.03) but not sEPSCs ($n_{+/+}$ = 15, $n_{+/-}$ = 13; $U$ = 87.0, p=0.65; $t_{26}$ = 0.19, p=0.85) was lower in $Nf1^{+/-}$ putative dopaminergic neurons. (G) Amplitude of sIPSCs ($n_{+/+}$ = 18, $n_{+/-}$ = 17; $U$ = 96.5, p=0.06; $t_{33}$ = 1.63, p=0.11) and sEPSCs ($n_{+/+}$ = 15, $n_{+/-}$ = 13; $U$ = 90.0, p=0.75; $t_{26}$ = 0.07, p=0.94). (H) 100 µM picrotoxin rescued spontaneous firing of $Nf1^{+/-}$ putative dopaminergic neurons ($n_{+/+}$ = 16, $n_{+/-}$ = 13; two-way ANOVA with Bonferroni *post hoc* tests; $F_{1,55}$ = 5.18, $p_{genotype \times drug}$ = 0.03; control: $p_{+/+ vs +/-}$ = 0.03, picrotoxin: $p_{+/+ vs +/-}$ > 0.99) relative to control neurons ($n_{+/+}$ = 12, $n_{+/-}$ = 18) and (I) lowered rheobase ($n_{+/+}$ = 25, $n_{+/-}$ = 20) relative to control $Nf1^{+/-}$ neurons ($n_{+/+}$ = 21, $n_{+/-}$ = 24; $F_{1,91}$ = 30.0, $p_{genotype \times drug}$ <0.001; control: $p_{+/+ vs +/-}$ < 0.001, picrotoxin: $p_{+/+ vs +/-}$ = 0.003, $Nf1^{+/-}$: $p_{control vs picrotoxin}$ <0.001). * denotes p<0.05 vs $Nf1^{+/+}$. # denotes p<0.05 vs control. Data presented as mean ± SEM, except box plots in F-G.

The online version of this article includes the following figure supplement(s) for figure 3:

**Figure supplement 1.** Effect of 20 µM forskolin (FSK) on putative dopaminergic neuron $I_h$ currents.

**Figure supplement 2.** Effect of 5 mg/kg morphine sulfate on spontaneous dLight1.2 transients in $Nf1^{+/-}$ mice.

Given these findings, we next sought to determine if pharmacological inhibition of VTA GABAergic neurons was sufficient to rescue dLight1.2 transient rates in $Nf1^{+/-}$ mice. µ-opioid receptor (MOR) agonists, such as morphine or DAMGO, robustly increase dopaminergic neuron firing and NAc dopamine release via pre-synaptic inhibition of GABAergic neurotransmission in the VTA (*Badiani et al., 2011*; *Di Chiara and Imperato, 1988*; *Johnson and North, 1992*). This model is supported by recent efforts by Lüscher and colleagues that combined NAc dLight1 monitoring and optogenetic manipulation of VTA sub-populations to examine cell-type-specific substrates of heroin (diacetylmorphine) reinforcement (*Corre et al., 2018*). We found that pre-treating $Nf1^{+/-}$ mice with the mu opioid receptor agonist morphine sulfate (5 mg/kg, s.c.) raised spontaneous dLight1.2 transient event rate but not event magnitude or FWHM relative to saline (*Figure 3—figure supplement 2*). This elevated event rate following MOR agonist exposure (0.54 ± 0.03 Hz) was not statistically

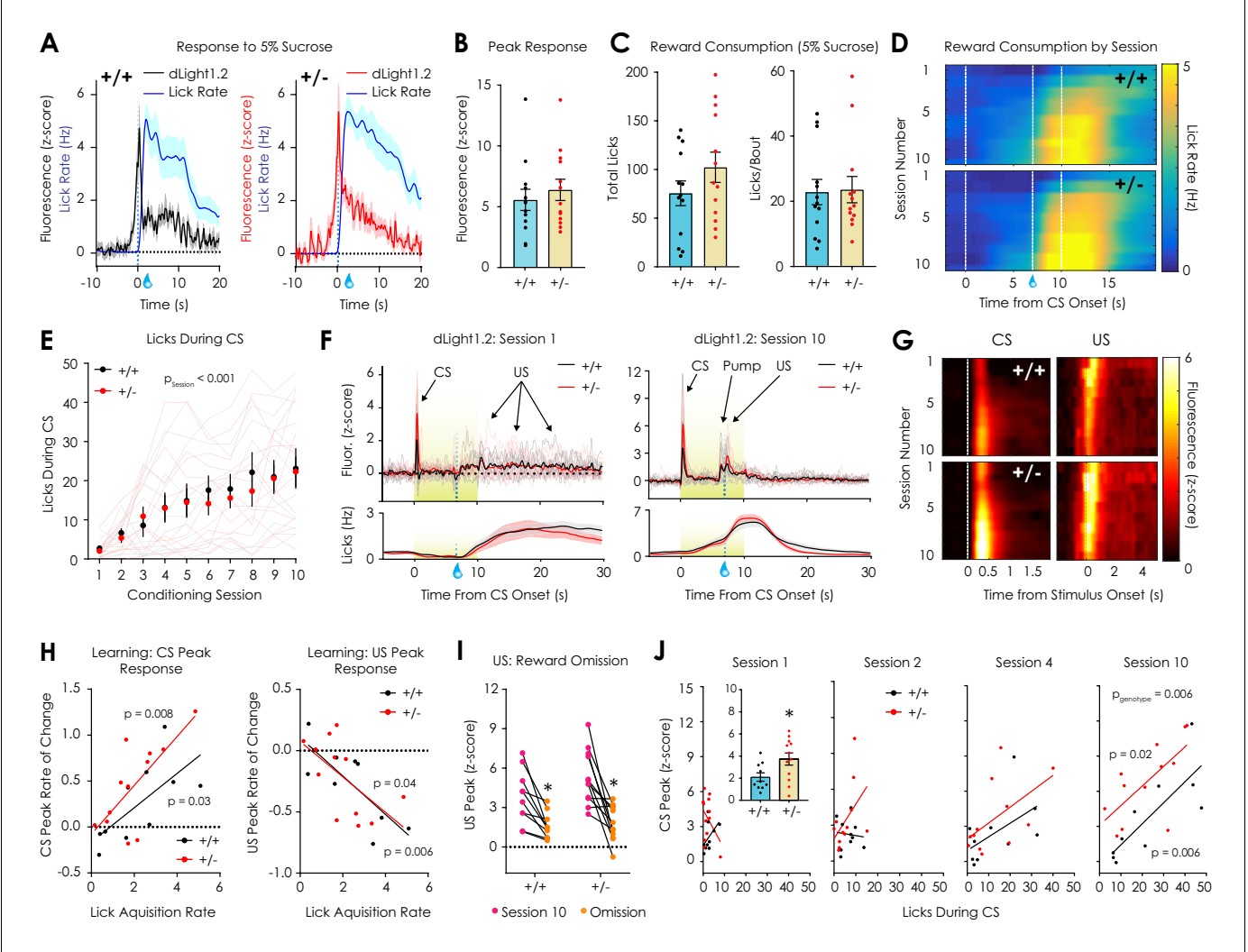

**Figure 4.** In vivo optical monitoring of dopamine dynamics during reward consumption and Pavlovian conditioning. (A) Consumption of 5% sucrose evoked robust, time-locked fluorescent dopamine transients in both $Nf1^{+/+}$ (*left*) and $Nf1^{+/-}$ mice (*right*). (B) Peak dLight1.2 responses to the onset sucrose consumption ($n_{+/+}$ = 13, $n_{+/-}$ = 13; unpaired t-test; $t_{24}$ = 0.66, p=0.51). (C) No difference in total number of licks (*left*; $t_{24}$ = 1.33, p=0.20) or licks per bout (*right*; $t_{24}$ = 0.14, p=0.89) were observed between genotypes. (D) Average session-by-session reward seeking during Pavlovian conditioning; the unconditioned stimulus (US, 5% sucrose) was delivered 7 s after the onset of a reward-predictive 10 s conditioned stimulus (CS, 5 kHz tone with house light illumination). (E) $Nf1^{+/+}$ (n = 10) and $Nf1^{+/-}$ (n = 12) mice displayed learned licking during the CS that was not dependent on genotype (two-way repeated measures ANOVA; $F_{9,180}$ = 0.48, $p_{genotype \times session}$ = 0.89; $F_{9,180}$ = 21.36, $p_{session}$ <0.001; $F_{1,20}$ = 0.09, $p_{genotype}$ = 0.77). (F) Individual averaged dLight1.2 traces before (*left*, Session 1) and after (*right*, Session 10) learning showing CS, US, and pump responses. (G) Heatmap showing average dLight1.2 responses to the CS (*left*; two-way repeated measures ANOVA; peak response: $F_{9,180}$ = 0.81, $p_{genotype \times session}$ = 0.61) or US (*right*; peak response: $F_{9,180}$ = 0.49, $p_{genotype \times session}$ = 0.88) across training sessions. (H) Across sessions, the rate of acquisition of licking during the CS was correlated with the rate of change of the CS ($Nf1^{+/+}$: $R^2$ = 0.48, p=0.03; $Nf1^{+/-}$: $R^2$ = 0.52, p=0.008) and US peak ($Nf1^{+/+}$: $R^2$ = 0.63, p=0.006; $Nf1^{+/-}$: $R^2$ = 0.36, p=0.03) in both genotypes. (I) Unexpected omission resulted in a significant reduction in US magnitude in both $Nf1^{+/+}$ (n = 10; paired t-test; $t_9$ = 4.03, p=0.003) and $Nf1^{+/+}$ mice (n = 12; paired t-test; $t_{11}$ = 4.50, p<0.001). (J) Correlation between CS peak response and CS licking during session 1 ($Nf1^{+/+}$: $R^2$ = 0.25, p=0.14; $Nf1^{+/-}$: $R^2$ = 0.16, p=0.21; $p_{genotype}$ = 0.04; *inset*: average peak; unpaired t-test; $t_{20}$ = 2.34, p=0.03), session 2 ($Nf1^{+/+}$: $R^2$ = 0.008, p=0.80; $Nf1^{+/-}$: $R^2$ = 0.22, p=0.12; $p_{genotype}$ = 0.14), session 4 ($Nf1^{+/+}$: $R^2$ = 0.28, p=0.12; $Nf1^{+/-}$: $R^2$ = 0.30, p=0.07; $p_{genotype}$ = 0.26), and session 10 ($Nf1^{+/+}$: $R^2$ = 0.63, p=0.006; $Nf1^{+/-}$: $R^2$ = 0.46, p=0.02; $p_{genotype}$ = 0.006). * denotes p<0.05. Data presented as mean ± SEM.

The online version of this article includes the following figure supplement(s) for figure 4:

**Figure supplement 1.** dLight1.2 responses to social interaction and measurement of social preference.
**Figure supplement 2.** Additional data: Pavlovian conditioning.

different (unpaired t-test; $t_{26}$ = 1.08, p=0.29) from the spontaneous event rate in $Nf1^{+/+}$ mice

(0.50 ± 0.02 Hz). Thus, excitation/inhibition imbalance is a mechanism gating $Nf1^{+/-}$ dopaminergic excitability ex vivo, and attenuation of VTA GABAergic neurotransmission normalizes spontaneous LNAc dopaminergic neurotransmission in vivo.

## Optical monitoring of dopamine responses to conditioned and unconditioned rewards

After measuring dLight1.2 signals at baseline and parsing these differences ex vivo, we next probed dopaminergic responses to salient stimuli. Because dopaminergic circuits respond strongly to rewards and reward-predictive cues (*Schultz et al., 2015*), we monitored LNAc dopamine signals in water-restricted mice during consumption of 5% sucrose. In both $Nf1^{+/-}$ and $Nf1^{+/+}$ mice, we observed robust LNAc dopamine transients time-locked to reward consumption (*Figure 4A*) that were not significantly different between genotypes (*Figure 4B*). No difference in the number of rewards consumed during the 30 min session was observed between groups (*Figure 4C*). We next measured the dLight1.2 response to social interaction, which is a positive reinforcer in mice (*Martin and Iceberg, 2015*). We observed large transients at the onset of interaction with a novel, sex-matched, juvenile conspecific that was independent of genotype (*Figure 4—figure supplement 1*). $Nf1^{+/-}$ and $Nf1^{+/+}$ littermates also failed to display differences in preference for a novel mouse in a social preference task (*Figure 4—figure supplement 1*). These findings suggest that LNAc dopamine and behavioral responses to unconditioned rewards are preserved in the context of $Nf1$ haploinsufficiency.

Dopaminergic populations have been widely studied for their role in reward learning (*Keiflin and Janak, 2015*; *Schultz et al., 2015*; *Wise, 2004*), so we optically monitored dopaminergic neurotransmission during a Pavlovian conditioning assay in water-restricted $Nf1^{+/+}$ and $Nf1^{+/-}$ mice. In this task, a 5% sucrose reward (the unconditioned stimulus or US) was delivered seven seconds after the beginning of a ten-second reward-predictive cue or conditioned stimulus (the CS; a 5 kHz tone with house light illumination) during ten, twenty-trial sessions. As each mouse learned the cue-reward association, the number of licks during CS presentation increased across sessions (*Figure 4D*). No differences in the number of licks during the CS, the number of anticipatory licks, or the learning rate (the slope of the linear fit of CS licks across trials) were observed between $Nf1^{+/+}$ and $Nf1^{+/-}$ mice (*Figure 4E*, *Figure 4—figure supplement 2*). In both genotypes, dLight1.2 peaks were observed in response to both CS presentation and US consumption (*Figure 4F*), and in later trials, to the sound of the sucrose delivery pump (*Figure 4—figure supplement 2*). Similar to previous studies (*Patriarchi et al., 2018*), dLight1.2 responses to the CS and US (*Figure 4G*) were enhanced and diminished, respectively, in mice that successfully learned the cue-reward association (i.e. the learning rate was correlated with the rate of change of each feature peak across trials; (*Figure 4H*, *Figure 4—figure supplement 2*). Unexpected omission of the US following CS presentation after learning equivalently diminished the dopamine response to reward seeking in both $Nf1^{+/-}$ and $Nf1^{+/+}$ mice (*Figure 4I*), which is consistent with the role of dopamine in reward prediction error detection (*Schultz et al., 2015*).

$Nf1^{+/-}$ mice displayed enhanced dLight1.2 responses to CS presentation on the first day of testing (*Figure 4J*, *inset*) compared to $Nf1^{+/+}$ littermates that was largest during the first trial (*Figure 4—figure supplement 2*). This phenotypic difference attenuated over subsequent days and re-emerged as the CS response became correlated with performance (*Figure 4J*, *Figure 4—figure supplement 2*). The magnitude of the CS response on the first day of testing did not predict task performance in later sessions (Session one vs Session 10; $Nf1^{+/+}$: $R^2$ = 0.02, p=0.69; $Nf1^{+/-}$: $R^2$ = 0.0006, p=0.94). Across sessions, a significant main effect of genotype on US magnitude was observed, although US magnitude was only greater in $Nf1^{+/-}$ mice during session 3 (*Figure 4—figure supplement 2*). No differences in response to the sucrose delivery pump, a purely auditory CS, were observed between genotypes across sessions (*Figure 4—figure supplement 2*). These findings indicate that, although CS responses are larger in $Nf1^{+/-}$ mice, the ability to form cue-reward associations is equivalent between genotypes and coincides with adaptive changes in dopaminergic neurotransmission with learning.

# Optical monitoring of dopaminergic neurotransmission during cued fear conditioning

Mesolimbic dopaminergic neurons are a heterogeneous population that exhibit diverse response profiles during exposure to aversive stimuli (*Ilango et al., 2012*; *de Jong et al., 2019*; *Lammel et al., 2014*), so we next recorded dopamine dynamics in a subset of mice undergoing cued fear conditioning. In this 15-trial assay, a 10 s audiovisual CS (house light and 3 kHz tone) predicted a 1 s foot shock (US), and the development of freezing during CS presentation was used as a proxy for learning. A previous study (*Silva et al., 1997*) failed to detect differences in cued fear conditioning between *Nf1*$^{+/+}$ and *Nf1*$^{+/-}$ mice, so we employed smaller shock (0.4 mA vs. 0.75 mA) and tone intensities (60 dB vs. 85 dB), shorter CS (10 s vs. 30 s) and US durations (1 vs. 2 s), and a repeated trial structure (15 trials vs one trial) in order to avoid a ceiling affect. Over the course of fifteen CS-US pairings, both *Nf1*$^{+/+}$ and *Nf1*$^{+/-}$ mice exhibited trial-by-trial increases in freezing during the first seven trials that subsequently plateaued (*Figure 5A*). In both genotypes, the expression of freezing in later trials (average freezing time, trials 8–15) was correlated with the acquisition rate (slope of the linear fit, trials 1–7; *Figure 5B*) and the latency to freeze (trials 8–15; *Figure 5—figure supplement 1*).

Compared to *Nf1*$^{+/+}$ littermates, both the acquisition rate and average freezing time was lower in *Nf1*$^{+/-}$ mice, which could be accounted for by an increased latency to freeze (*Figure 5C*). Qualitative review of behavioral video recordings revealed that presentation of the CS resulted in several seconds of locomotor stimulation in *Nf1*$^{+/-}$ mice that delayed freezing (*Video 1*), whereas *Nf1*$^{+/+}$ mice froze with short latency at CS onset once learning had occurred (*Video 2*). During LNAc dLight1.2 monitoring, a dopamine transient was observed at the onset of CS presentation that was greatest in trial 1, attenuated across trials, and was larger in *Nf1*$^{+/-}$ mice (*Figure 5D–E*). In *Nf1*$^{+/-}$ but not *Nf1*$^{+/+}$ mice, the magnitude of the CS peak during trial one was negatively correlated with acquisition rate and freezing duration and positively correlated with latency to freeze (*Figure 5F–G*, *Supplementary file 1*). During US (shock) delivery, we observed an initial positive dopamine transient followed by a 1 to 2 s negative anti-peak and a subsequent, broader post-US rebound that returned to baseline several seconds later (*Figure 5D,H*). This waveform mirrors patterns of activity observed during extracellular recordings in the VTA (*Brischoux et al., 2009*) and GCaMP monitoring of dopaminergic axons in the LNAc (*de Jong et al., 2019*).

During US exposure, *Nf1*$^{+/-}$ mice had significantly larger initial US peak responses and a smaller integrated post-US rebound compared to *Nf1*$^{+/+}$ littermates (area under the curve; *Figure 5H*, *Figure 5—figure supplement 1*). US anti-peak and post-US rebound peak responses were equivalent between genotypes across trials (*Figure 5H*, *Figure 5—figure supplement 1*). In *Nf1*$^{+/+}$ mice, the magnitude of the US anti-peak during trial one was negatively correlated with freezing and positively correlated with latency to freeze (*Figure 5I–J*, *Supplementary file 2*). Additionally, the magnitude of the integrated post-US rebound was negatively correlated with the latency to freeze in *Nf1*$^{+/-}$ mice (*Figure 5I–J*, *Supplementary file 2*). These findings demonstrate that *Nf1*$^{+/-}$ mice exhibit altered patterns of dopaminergic neurotransmission in response to both shock-predictive cues and shock delivery, which correlated with behavioral responses during the task. Larger dopaminergic responses to the onset of the CS in the first trial was associated with longer latencies to freeze and shorter freezing durations in *Nf1*$^{+/-}$ mice, while more negative dopamine responses to shock delivery during trial one was predictive of shorter latencies to freeze and longer freezing durations in *Nf1*$^{+/+}$ mice.

## Dopaminergic and behavioral responses to salient visual stimuli

In order to investigate the etiology of the dopaminergic response to the CS, we measured dLight1.2 responses to either a 10 s overhead light or a 3 kHz tone (inter-trial interval: 75–90 s) randomly presented during a 20-trial session.

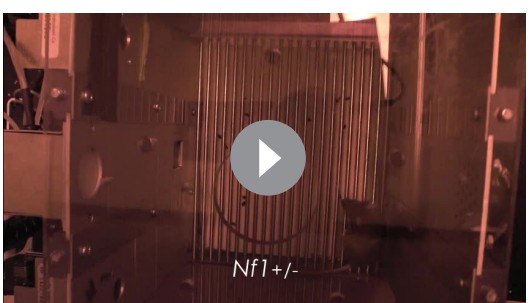

**Video 1.** Behavioral response of *Nf1*$^{+/-}$ mouse to CS presentation during fear conditioning.
https://elifesciences.org/articles/48983#video1

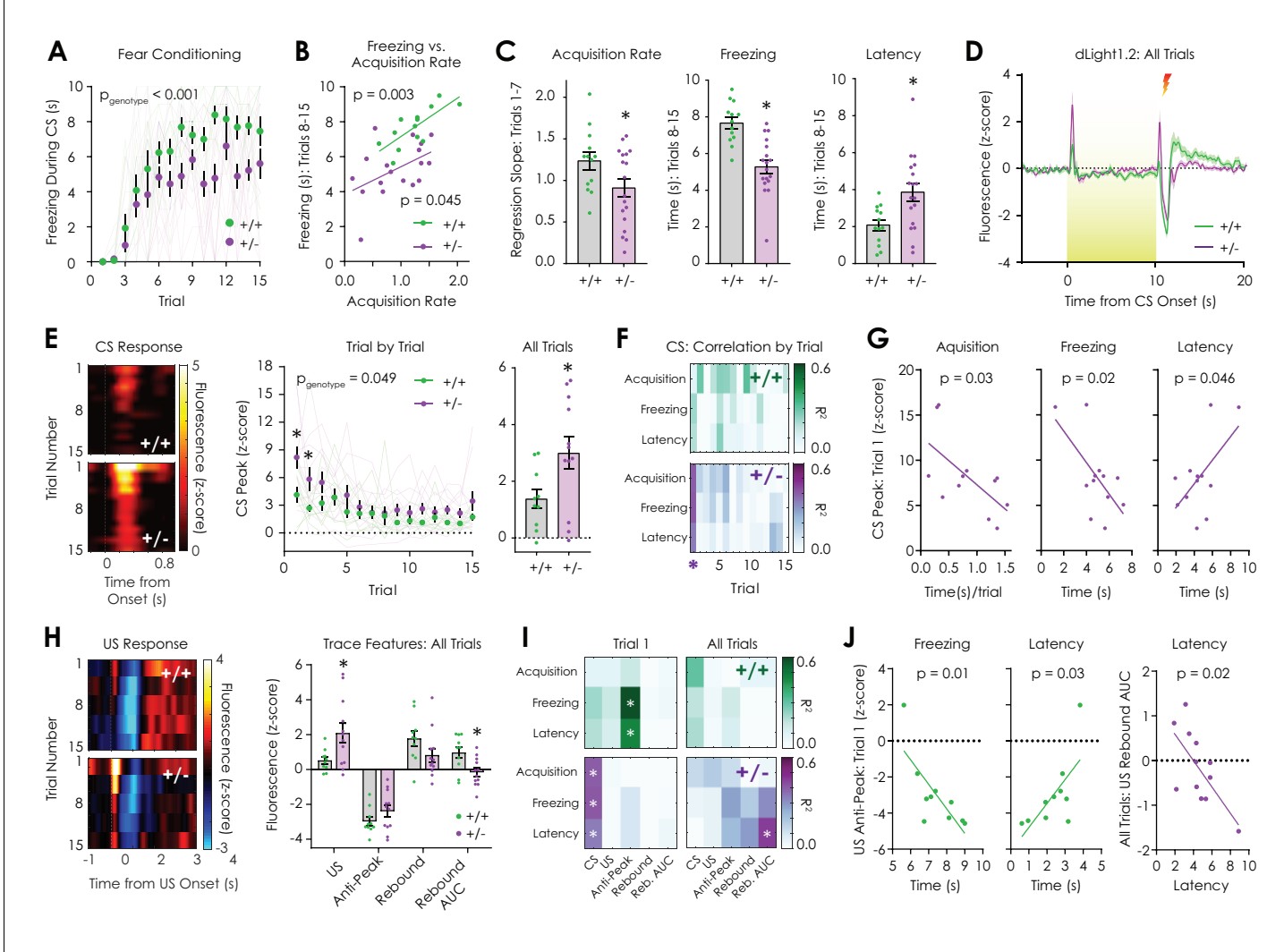

**Figure 5.** In vivo optical monitoring of dopamine dynamics during cued fear conditioning. (A) During the cued fear conditioning assay, mice displayed a trial-by-trial increase in freezing that was greater in *Nf1+/+* mice but not dependent on genotype ($n_{+/+}$ = 13, $n_{+/-}$ = 18; 2-way repeated measures ANOVA; $F_{14,406}$ = 1.321, $p_{genotype \times trial}$ = 0.19; $F_{14,406}$ = 28.56, $p_{trial}$ <0.001; $F_{1,29}$ = 18.54, $p_{genotype}$ <0.001). (B) The freezing acquisition rate during trials 1–7 was correlated with freezing during trials 8–15 in both *Nf1+/+* (n = 13; $R^2$ = 0.56, p=0.003) and *Nf1+/-* mice (n = 18; $R^2$ = 0.23, p=0.045). (C) The freezing acquisition rate (*left*; unpaired t-test; $t_{29}$ = 2.08, p=0.046) and average freezing during trials 8–15 (*middle*; $t_{29}$ = 4.79, p<0.001) were decreased in *Nf1+/-* mice due to increased latency to freeze (*right*; $t_{29}$ = 2.90, p=0.007). (D) Averaged dLight1.2 traces showing responses to CS (10 s, 3 kHz tone with house light illumination) presentation and US (1 s, 0.4 mA shock) delivery. (E) Heatmaps showing trial-by-trial changes in dLight1.2 signal in response to the CS (*left*). *Nf1+/-* mice (n = 12) displayed increased CS responses across trials (*middle*; two-way repeated measures ANOVA; $F_{14,280}$ = 1.662, $p_{genotype \times trial}$ = 0.06; $F_{14,280}$ = 9.30, $p_{trial}$ <0.001; $F_{1,20}$ = 4.37, $p_{genotype}$ = 0.049) and when traces were averaged (*right*; unpaired t-test; $t_{20}$ = 2.324, p=0.03) compared to *Nf1+/+* mice (n = 10). (F) Correlation matrix showing trial-by-trial correlation strength between behavioral measures and CS peak response. (G) In *Nf1+/-* mice, there were significant correlations between the CS peak in trial one and the freezing acquisition rate ($R^2$ = 0.40, p=0.03), time spent freezing ($R^2$ = 0.41, p=0.02), and the latency to freeze ($R^2$ = 0.34, p=0.046). (H) Heatmaps showing trial-by-trial changes in dLight1.2 signal in response to US delivery (*left*). *Nf1+/-* mice (n = 12) exhibited increased average peak responses (*right*) to US onset ($t_{20}$ = 2.50, q = 0.04) and decreased integrated post-US rebound (area under the curve or AUC; $t_{20}$ = 2.85, q = 0.03) compared to *Nf1+/+* mice (n = 10). (I) Correlation matrices displaying strength of US feature peak-behavior correlations during trial one and across trials. (J) There were significant correlations between the US anti-peak magnitude in trial one and freezing ($R^2$ = 0.58, p=0.01) or the latency to freeze ($R^2$ = 0.46, p=0.03) in *Nf1+/+* mice and the integrated post-US rebound across all trials and the latency to freeze ($R^2$ = 0.45, p=0.02) in *Nf1+/-* mice. *denotes p<0.05. Multiple t-tests were corrected with the two-stage linear step-up procedure of Benjamini, Krieger, and Yekutieli with a false discovery rate of 5%. Data presented as mean ± SEM.

The online version of this article includes the following figure supplement(s) for figure 5:

**Figure supplement 1.** Additional data: Cued fear conditioning.

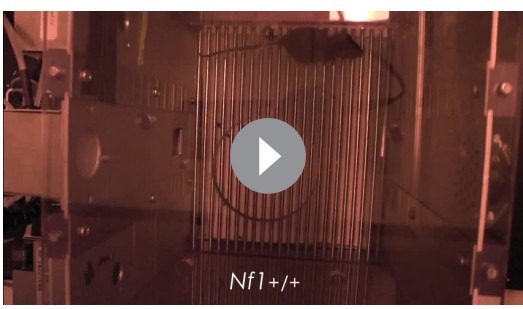

**Video 2.** Behavioral response of *Nf1*^+/+ mouse to CS presentation during fear conditioning.
https://elifesciences.org/articles/48983#video2

Both genotypes exhibited robust dopamine transients at the onset of the overhead light stimulus that returned to baseline within 1–2 s (*Figure 6A*) and decremented across trials (*Figure 6—figure supplement 1*). In both *Nf1*^+/+ mice and *Nf1*^+/- mice, 60 dB, 3 kHz auditory tone evoked dopamine transients at stimulus onset (*Figure 6B*) that were comparatively smaller than light responses (*Figure 6C*) and non-trial-dependent (*Figure 6—figure supplement 1*). *Nf1*^+/- mice exhibited larger responses to light but not tone onset (*Figure 6C*), which was confirmed when traces were analyzed as ΔF/F rather than z-score (*Figure 6—figure supplement 1*). These findings raise the possibility that phenotypic differences in dopaminergic CS responses are driven by reactions to an overhead visual stimulus. In order to investigate if the overhead light affected performance during cued fear conditioning, we performed the assay using a tone-only CS (3 kHz tone). During the last five trials, tone-only trials were interleaved with trials in which the house light was added to the CS (trials 11, 13, 15). We found that the development of cued freezing in *Nf1*^+/- mice was equivalent to *Nf1*^+/+ littermates across the first ten trials (*Figure 6D*). Addition of the overhead light stimulus to the CS was sufficient to perturb the expression of freezing in *Nf1*^+/- but not *Nf1*^+/+ mice by increasing the latency to freeze (*Figure 6E*, *Video 3*). Thus, deficits in cued fear conditioning in *Nf1*^+/- mice are reversible, visual stimulus-dependent, and independent of learning.

Dopaminergic neurons, in addition to their role in processing rewarding or aversive outcomes, respond to salient alerting signals to modulate attentional orientation and promote appropriate motivated responses (*Bromberg-Martin et al., 2010*; *Schultz and Romo, 1990*). Because enhanced dopaminergic responses to an overhead light may reflect increased motivational salience of an alerting visual stimulus (*Thompson et al., 2010*), we performed a looming stimulus assay in *Nf1*^+/+ and *Nf1*^+/- mice. During this test, subjects are exposed to an expanding overhead disc designed to mimic predator approach that rapidly promotes escape and/or freezing behaviors in rodents (*Yilmaz and Meister, 2013*). In response to the onset of the looming stimulus, both genotypes exhibited similar sub-second reaction times (*Figure 6—figure supplement 2*). Compared to *Nf1*^+/+ littermates, *Nf1*^+/- mice were more likely to escape to the shelter at stimulus onset and exhibited shorter latencies to the first freezing episode (*Figure 6F*). In both groups, freezing latency was not dependent on freezing location (*Figure 6—figure supplement 2*). No differences in the length of the first freezing episode or total freezing during the first minute after looming were observed between genotypes, although freezing was dependent on escape location: mice that escaped to the shelter exhibited shorter freezing durations than those in the open field (*Figure 6—figure supplement 2*).

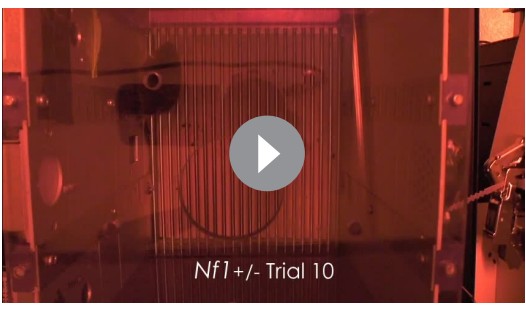

**Video 3.** Behavioral response of *Nf1*^+/- mouse to CS presentation during tone-only and interleaved light trials.

https://elifesciences.org/articles/48983#video3

Recently, it has been shown that flight-to-shelter responses to looming stimuli are mediated by VTA GABAergic neurons and driven by excitatory projections from the ventral superior colliculus (vSC) (*Zhou et al., 2019*). This projection innervates both dopaminergic and GABAergic neurons in the VTA (*Prévost-Solié et al., 2019*; *Zhou et al., 2019*) and additionally regulates orientation during social interaction (*Prévost-Solié et al., 2019*). In order to determine if the vSC can induce dopaminergic neurotransmission in a manner similar to overhead light exposure, we stereotaxically injected an AAV vector (AAV5-hSyn-ChR2(H134R)-eYFP) into the vSC to express the light-gated ion channel channelrhodopsin-2 (ChR2) in neurons, followed by implantation of an

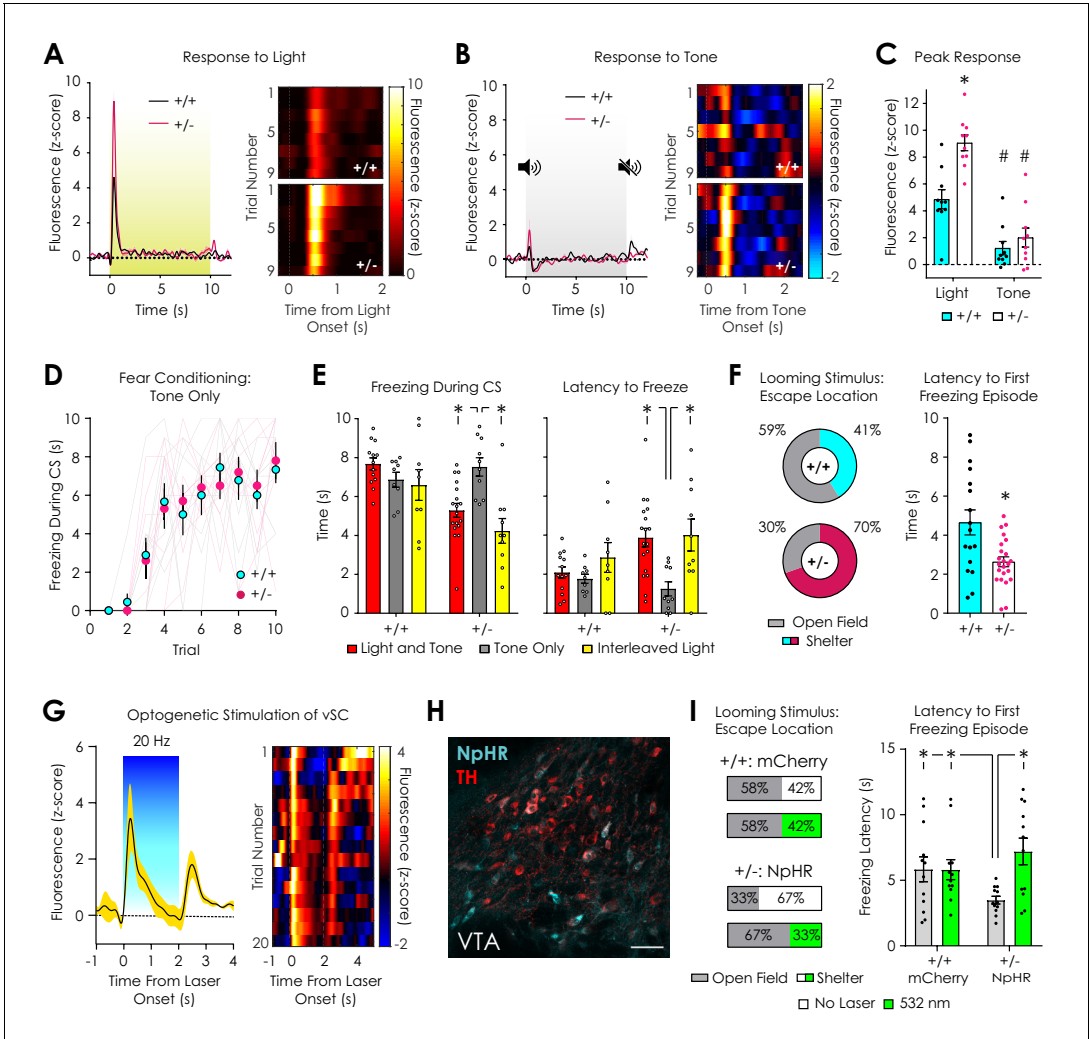

**Figure 6.** Dopaminergic and behavioral responses to salient visual stimuli. (**A**) Average (*left*) and trial-by-trial (*right*) fluorescent dopamine response to a 10 s overhead light stimulus. (**B**) Average (*left*) and trial-by-trial (*right*) fluorescent dopamine response to a 10 s auditory stimulus (5 kHz tone). (**C**) *Nf1*$^{+/-}$ mice had greater peak responses to light (p<0.001) but not tone onset (p>0.99) compared to *Nf1*$^{+/+}$ mice ($n_{+/+}$ = 10, $n_{+/-}$ = 10; two-way ANOVA with Bonferroni post hoc tests; $F_{1,36}$ = 7.27, $p_{genotype \times condition}$ = 0.01; $F_{1,36}$ = 15.48, $p_{genotype}$ <0.001). In both genotypes, responses to light were greater than responses to tone ($F_{1,36}$ = 71.02, $p_{stimulus}$ <0.001; $p_{+/+}$ = 0.002, $p_{+/-}$ < 0.001). (**D**) No difference in cued fear conditioning was observed when a tone-only CS was used ($n_{+/+}$ = 9, $n_{+/-}$ = 10; two-way repeated measures ANOVA; $F_{9,153}$ = 0.26, $p_{genotype \times trial}$ = 0.98). (**E**) *Nf1*$^{+/-}$ mice exhibited increased freezing (*left*) and decreased latency to freeze (*right*) in tone-only CS trials (n = 10) compared to light and tone (n = 18; unpaired t-test; freezing: $t_{26}$ = 3.75, p<0.001; latency: $t_{26}$ = 3.75, p<0.00) or interleaved light trials (n = 10; paired t-test; freezing: $t_9$ = 5.30, p<0.001; latency: $t_9$ = 3.48, p=0.007). No differences in freezing or latency to freeze was observed between tone-only CS trials (n = 9) and light and tone (n = 13; unpaired t-test; freezing: $t_{20}$ = 1.66, p=0.11; latency: $t_{20}$ = 0.81, p=0.43) or interleaved light trials (n = 10; paired t-test; freezing: $t_8$ = 0.42, p=0.69; latency: $t_8$ = 1.49, p=0.19) in *Nf1*$^{+/+}$ mice. (**F**) *Nf1*$^{+/+}$ (n = 17) and *Nf1*$^{+/-}$ mice (n = 23) had similar reaction times to a looming stimulus (*left*; $t_{38}$ = 0.79, p=0.43), yet *Nf1*$^{+/+}$ mice were more likely to escape to the shelter after stimulus presentation (*left*) and exhibited shorter latency to the first freezing episode after looming onset than *Nf1*$^{+/-}$ mice ($t_{38}$ = 3.24, p=0.003). (**G**) Optogenetic stimulation of the ventral superior colliculus (vSC) produced time-locked dopamine release in the LNAc (n = 3 mice; average trace, *left*; trial-by-trial response, *right*). (**H**) Representative confocal image showing tyrosine hydroxylase (TH)-positive dopaminergic and *Th*-Off-NpHR-eYFP neurons in the VTA (scale: 50 µm). (**I**) In the absence of photoinhibition, VTA$^{Th\text{-}Off\text{-}NpHR\text{-}eYFP}$ *Nf1*$^{+/-}$ mice (n = 12) were more likely to escape to the shelter (*left*) and had shorter latency to the first freezing episode (*right*; unpaired t-test; $t_{22}$ = 2.36, p=0.03) compared with VTA$^{Th\text{-}Off\text{-}mCherry}$ *Nf1*$^{+/+}$ mice (n = 12). Optogenetic inhibition of VTA$^{non\text{-}Th}$ neurons with 532 nm light (5 mW, 30 Hz, 20 ms pulse width) decreased the probability of escape to the shelter (*left*) and increased the latency to the first freezing episode (*right*; paired t-test; $t_{11}$ = 3.82, p=0.003) in VTA$^{Th\text{-}Off\text{-}NpHR\text{-}eYFP}$ *Nf1*$^{+/-}$ mice to levels that were similar to VTA$^{Th\text{-}Off\text{-}mCherry}$ *Nf1*$^{+/+}$ mice (unpaired t-test; *Nf1*$^{+/-\ Laser\ On}$ vs *Nf1*$^{+/+\ Laser\ Off}$: $t_{22}$ = 0.98, p=0.34; *Nf1*$^{+/-\ Laser\ On}$ vs *Nf1*$^{+/+\ Laser\ On}$: $t_{22}$ = 1.09, p=0.29). No difference was observed in VTA$^{Th\text{-}Off\text{-}mCherry}$ *Nf1*$^{+/+}$ mice between stimulation conditions (paired t-test; $t_{11}$ = 0.02, p=0.99). *denotes p<0.05. # denotes p<0.05 vs light stimulus (panel C). Data presented as mean ± SEM.

The online version of this article includes the following figure supplement(s) for figure 6:

**Figure supplement 1.** Additional data: dLight1.2 responses to auditory and visual stimuli.

*Figure 6 continued on next page*

*Figure 6 continued*

**Figure supplement 2.** Additional data: Looming stimulus assay.
**Figure supplement 3.** Additional data: Optogenetic control of the vSC and VTA$^{non-Th}$ neurons.

optical fiber for 473 nm light delivery in C57Bl/6N mice. A photometry fiber was also implanted in the ipsilateral LNAc for simultaneous dLight1.2 monitoring during vSC photostimulation. Similar to overhead light exposure, two seconds of vSC optogenetic stimulation (5 mW, 20 Hz, 5 ms pulse width) produced time-locked, short latency dopamine release in the LNAc at stimulus onset that subsequently decayed to baseline (*Figure 6G*, *Figure 6—figure supplement 3*). In early trials, the signal decayed to sub-baseline, negative values. Discontinuation of the laser was associated with a robust dopaminergic rebound. During testing, we also observed that vSC ChR2 activation evoked vigorous escape behavior in the absence of a threatening stimulus (*Video 4*), as has been observed following photostimulation of vSC-to-VTA projections (*Zhou et al., 2019*). These findings indicate that the SC, a visually responsive region involved in innate responses to salient visual stimuli (*Ito and Feldheim, 2018*), can induce rapid dopamine release in the LNAc that is qualitatively similar to overhead light exposure.

## Optogenetic inhibition of non-dopaminergic neurons in the VTA of *Nf1$^{+/-}$* mice during looming stimulus presentation

Optogenetic inhibition of VTA GABAergic neurons or vSC projections in the VTA is sufficient to suppress looming stimulus responses in mice (*Zhou et al., 2019*). Given that *Nf1* heterozygosity is associated with excitation-inhibition imbalance in the VTA, we hypothesized that suppression of inhibitory neurotransmission during looming stimulus presentation would normalize behavioral responses in *Nf1$^{+/-}$* mice. To test this hypothesis, we used an intersectional strategy to target GABAergic neurons in the VTA for optogenetic silencing with the light-gated chloride pump halorhodopsin (NpHR) during behavior. We co-injected an AAV vector to express Cre recombinase under control of the *Th* promoter in dopaminergic neurons (AAV9-*Th*-PI-Cre-SV40) with either a Cre-Off NpHR (AAV-DJ-Ef1α-DO-eNpHR3.0-eYFP, *Nf1$^{+/-}$* mice) or mCherry (AAV-DJ-Ef1α-DO-mCherry, *Nf1$^{+/+}$* mice) vector bilaterally into the VTA (*Saunders et al., 2012*), followed by implantation of 300 μm optical fibers for light delivery. Because the mCherry or NpHR transgenes were double-floxed with an open reading frame (DO), they would be inactivated in the presence of Cre but express normally in non-dopaminergic neurons (*Figure 6H*). Given that ~90% of non-dopaminergic neurons are GABAergic in the VTA (*Morales and Root, 2014*; *Pignatelli and Bonci, 2015*), we reasoned that our approach would provide efficient optogenetic control of inhibitory neurotransmission in this region. Post hoc analysis revealed that only 14.7 ± 0.01% of NpHR-positive cells in the VTA expressed tyrosine hydroxylase (*Figure 6H*, *Figure 6—figure supplement 3*), indicating successful targeting of VTA$^{non-Th}$ neurons in *Nf1$^{+/-}$* and *Nf1$^{+/+}$* mice.

In the absence of optogenetic inhibition, optical patch cable-tethered VTA$^{Th-Off-NpHR-eYFP}$ *Nf1$^{+/-}$* mice were more likely to escape to the available shelter in response to looming stimulus presentation and exhibited shorter latency to the first freezing episode when compared to VTA$^{Th-Off-mCherry}$ *Nf1$^{+/+}$* mice (*Figure 6I*). Delivery of 532 nm light (5 mW, 30 Hz, 20 ms pulse width) to the VTA during looming stimulus presentation had no effect on VTA$^{Th-Off-mCherry}$ *Nf1$^{+/+}$* mice but decreased the percentage of VTA$^{Th-Off-NpHR-eYFP}$ *Nf1$^{+/-}$* mice that escaped to the shelter to a level (33%) that was similar to *Nf1$^{+/+}$* mice (41%) and VTA$^{Th-Off-mCherry}$ *Nf1$^{+/+}$* mice (42%) (*Figure 6F–I*; *Video 5*). This shift in escape location coincided with an increase in the latency to the first freezing episode in VTA$^{Th-Off-NpHR-eYFP}$ *Nf1$^{+/-}$* mice that was not significantly different from VTA$^{Th-Off-mCherry}$ *Nf1$^{+/+}$* mice (*Figure 6I*). Thus, optogenetic inhibition of VTA$^{non-Th}$ neurons normalizes looming stimulus responses in *Nf1$^{+/-}$* mice and supports a role for VTA GABAergic neurons in the expression of visual stimulus response phenotypes caused by *Nf1* haploinsufficiency.

## Discussion

Developmental perturbations in mesencephalic dopaminergic circuits have been hypothesized to contribute to neurocognitive symptoms in NF1, yet their activity has never been directly investigated

in vivo. Here we leveraged the novel dopamine sensor dLight1.2 to assay mesoaccumbal dopamine dynamics in awake, behaving NF1 model mice and observed that the frequency of spontaneous fluorescent dopaminergic transients was lower in $Nf1^{+/-}$ mice. Using patch clamp electrophysiology, we showed that $Nf1^{+/-}$ dopaminergic neurons are less excitable and have lower spontaneous firing rates ex vivo due to increased GABAergic tone. Pharmacological or optogenetic inhibition of VTA GABAergic neurons rescued spontaneous dopaminergic and behavioral phenotypes, respectively. Given these findings, it is likely that increased tonic inhibition of $Nf1^{+/-}$ dopaminergic neurons reduces basal activity, whereas smaller soma volume acts as a compensatory mechanism to increase excitability during disinhibition. This would serve to facilitate bursting in response to a strong stimulus and maintain dLight1.2 transient amplitude, although future studies monitoring dopaminergic neuron activity with single cell resolution will be required to fully parse this hypothesis. Excitation/ inhibition imbalance is present in the amygdala (*Molosh et al., 2014*; *Repunte-Canonigo et al., 2015*), striatum (*Shilyansky et al., 2010*), and medial prefrontal cortex (mPFC) (*Gonçalves et al., 2017*; *Shilyansky et al., 2010*) of $Nf1^{+/-}$ mice and likely contributes to deficits in working memory, contextual fear conditioning, and social memory (*Cui et al., 2008*; *Molosh et al., 2014*; *Shilyansky et al., 2010*). Subthreshold doses of picrotoxin (0.01 mg/kg) (*Cui et al., 2008*) or L-DOPA (*Wozniak et al., 2013*) improve cognitive performance of NF1 model mice, emphasizing the therapeutic potential of interventions that modulate mesolimbic dopaminergic circuit function.

Although $Nf1^{+/-}$ dopaminergic neurons exhibit decreased soma size, we failed to detect changes in VTA or NAc TH immunofluorescence, as well as NAc monoamine content. This is in contrast with the optic glioma (OPG) mouse model of NF1 that has reduced tyrosine hydroxylase expression in the VTA (*Brown et al., 2012*; *Diggs-Andrews et al., 2013*) and lower TH, dopamine, and phosphorylated DARPP-32 (dopamine and cAMP-regulated phosphoprotein-32) in terminal fields (*Brown et al., 2010a*; *Diggs-Andrews et al., 2013*; *Anastasaki et al., 2015*). Neurite outgrowth and growth cone areas are decreased in cultured OPG dopaminergic neurons (*Diggs-Andrews et al., 2013*), yet we did not observe changes in neurite morphology in *Th*-VAST-labeled $Nf1^{+/-}$ neurons relative to $Nf1^{+/+}$ littermates. Incongruence between mouse models may be due to relative differences in neurofibromin expression. $Nf1^{+/-}$ mice on a pure C57Bl/6 background have higher tissue neurofibromin levels compared to OPG mice, and neurofibromin dose-dependently regulates TH and cellular dopamine production in cultured mouse neurons and neural progenitor cells differentiated from patient-derived induced pluripotent stem cells (*Anastasaki et al., 2015*). While the etiology of reduced soma volumes is unknown, cAMP deficiency influences the morphology of $Nf1^{+/-}$ and OPG neurons in vitro (*Brown et al., 2012*; *Diggs-Andrews et al., 2013*). Perturbations in this pathway thus represents a mechanism of interest in future efforts to characterize mesencephalic development in the context of NF1.

We also observed that $Nf1^{+/-}$ mice exhibit more robust dopaminergic responses to an overhead visual light stimulus, which was correlated with delayed freezing in a cued fear conditioning task. These findings are not indicative of an inability to form cue-outcome associations, since $Nf1^{+/-}$ mice performed similarly to $Nf1^{+/+}$ littermates when conditioning was carried out in the absence of the light stimulus. NAc dopaminergic neurotransmission is necessary for the acquisition and expression of learned fear responses (*Fadok et al., 2010*; *Fadok et al., 2009*), so enhanced CS responses would be expected to promote not attenuate conditioned fear. More likely, exaggerated dopaminergic responses to light presentation reflect increased stimulus salience, as overhead light stimuli are aversive in mice (*Thompson et al., 2010*), unexpected salient visual cues promote dopaminergic firing (*Bromberg-Martin et al., 2010*), and $Nf1^{+/-}$ mice had more robust behavioral responses to a looming stimulus. Work by Schultz and colleagues suggests that alerting dopamine signals, unlike reward responses, enable orientation and stimulus investigation (*Schultz, 2010*). As such, increased visual cue salience in $Nf1^{+/-}$ mice may have increased freezing latency during fear conditioning by provoking an investigative locomotor response in the absence of an obvious escape location. Because the looming disc is designed to simulate a predatory environmental cue, it has a greater negative valance; in this case, increased stimulus salience would promote flight-to-shelter responses and reduce the latency to the first freezing episode. Thus, bidirectional effects of $Nf1$ heterozygosity on freezing latency could be consistent with a single phenotypic process and depend on visual stimulus intensity and the ability to escape.

Adult and adolescent human subjects with NF1 exhibit visual processing deficits (*Hyman et al., 2005*), although visual cortical areas have sparse, if any, direct connections to VTA dopaminergic

neurons (*Watabe-Uchida et al., 2012*). Short latency dopaminergic responses to visual stimuli are likely driven by afferents from the superior colliculus (SC) (*Redgrave et al., 2010*), which receives direct input from retinal ganglion cells (*Dhande and Huberman, 2014*), responds to looming stimuli (*Zhao et al., 2014*), and evokes firing of both VTA GABAergic and dopaminergic neurons in vivo while enhancing flight-to-shelter responses (*Prévost-Solié et al., 2019*; *Zhou et al., 2019*). Because the dopaminergic response to optogenetic vSC stimulation occurred at stimulus onset, quickly attenuated, and was followed by a post-stimulation rebound, it is likely that excitatory vSC-to-VTA projections excite and subsequently suppress dopaminergic outflow via feed-forward inhibition, producing rebound disinhibition at stimulus offset. While the initial dopaminergic peak may serve as a salience signal, flight-to-shelter responses appear to be mediated by VTA GABA neurons (*Zhou et al., 2019*). In this case, increased vSC-to-VTA excitatory drive in $Nf1^{+/-}$ mice would be predicted to enhance LNAc dopamine release at stimulus onset, while subsequently driving signal termination and escape responses via GABAergic neurons. Thus, the role of tectal inputs to the VTA in moderating visual stimulus sensitivity in NF1 mouse models represents a promising focus for future efforts to parse disease symptomatology and may provide new insights into visual processing in patient populations.

# Materials and methods

## Key resources table

| Reagent type (species) or resource | Designation | Source or reference | Identifiers | Additional information |
|---|---|---|---|---|
| Antibody | Anti-tyrosine hydroxylase (Rabbit polyclonal) | EMD Millipore | Cat#: AB152 RRID:AB_390204 | IHC (1:1000) |
| Antibody | Anti-tyrosine hydroxylase (mouse monoclonal) | ImmunoStar | Cat#: 22941 RRID:AB_572268 | IHC (1:1000) |
| Antibody | Anti-GFP (mouse polyclonal) | Aves | Cat#: GFP-1020 RRID:AB_10000240 | IHC (1:1000) |
| Antibody | Alexa Fluor 488-conjugated donkey anti-chicken IgY F(ab')two fragment | Jackson Immuno Research | Cat#: 703-546-155 RRID: AB_2340376 | IHC (1:1000) |
| Antibody | Alexa Fluor 647-conjugated donkey anti-mouse IgG Fab fragment | Jackson Immuno Research | Cat#: 711-607-003 RRID: AB_2340626 | IHC (1:1000) |
| Recombinant DNA reagent | pAAV-hSyn-dLight1.2 | Addgene | Plasmid#: 111068 RRID: Addgene_111068 | Gift on Lin Tian; produced by UC Davis Vector Core |
| Recombinant DNA reagent | pAAV-hSyn-hChR2(H134R)-EYFP | Addgene | Plasmid#: 26973 RRID:Addgene_26973 | Gift of Karl Deisseroth; produced by UNC Vector Core |
| Recombinant DNA reagent | AAV9-Th-PI-Cre-SV40 | Addgene | Plasmid#: 107788 RRID: Addgene_107788 | Addgene viral prep#: 107788-AAV9; gift of James M. Wilson |

*Continued on next page*

*Continued*

| Reagent type (species) or resource | Designation | Source or reference | Identifiers | Additional information |
|---|---|---|---|---|
| Recombinant DNA reagent | pAAV-DJ-Ef1α-DO-eNpHR3.0-eYFP-WPRE-pA | Addgene | Plasmid#: 37087 RRID: Addgene_37087 | Gift of Bernardo Sabatini |
| Recombinant DNA reagent | pAAV-DJ-Ef1α-DO-mCherry-WPRE-pA | Addgene | Plasmid#: 37119 RRID: Addgene_37119 | Gift of Bernardo Sabatini |
| Recombinant DNA reagent | pAAV-ihSyn1-tTA-WPRE | Addgene | Plasmid#: 99120 RRID: Addgene_99120 | |
| Recombinant DNA reagent | pAAV-*Th*-tTA-WPRE | Addgene | Plasmid#: 133268 RRID: Addgene_133268 | |
| Recombinant DNA reagent | pAAV-TRE-mRuby-WPRE | Addgene | Plasmid#: 99114 RRID: Addgene_99114 | |
| Recombinant DNA reagent | pAAV-TRE-mNeonGreen-WPRE | (*Chan et al., 2017*) | | |
| Recombinant DNA reagent | pAAV-TRE-mTurquoise-WPRE | Addgene | Plasmid#: 99113 RRID: Addgene_99113 | |
| Recombinant DNA reagent | pAAV-*Th*-GFP-WPRE | Addgene | Plasmid#: 99128 RRID: Addgene_99128 | |
| Recombinant DNA reagent | pUCmini-iCAP-PHP.eB | Addgene | Plasmid#: 103005 RRID: Addgene_103005 | |
| Recombinant DNA reagent | pAAV-DJ-Rep-Cap | Cell Biolabs, Inc | Cat#: VPK-420-DK | |
| Software, Algorithm | Matlab | Mathworks, Inc | RRID:SCR_001622 | |
| Software, Algorithm | GraphPad Prism 7 | GraphPad Software, Inc | RRID:SCR_002798 | |
| Software, Algorithm | ABET II Software for Operant Control | Lafayette Instrument Company | Model 89501 | |
| Software, Algorithm | Fiber Photometry Trace Processing | Gradinaru Lab | FP_Session_Processing_2 .m | https://github.com/GradinaruLab/dLight1/blob/master/FP_Session_Processing2.m |
| Other | ProLong Diamond Antifade Mountant | ThermoFisher Scientific | Cat#: P36965 | |
| Other | Refractive Index Matching Solution | (*Yang et al., 2014*) | | Refractive Index = 1.46; protocol available in *Treweek et al. (2015)* |
| Other | Mono Fiber-Optic Cannula | Doric Lenses, Inc | Cat#: MFC_400/430–0.48_5 mm_ZF1.25_FLT | OD: 400 μm, Length: 5 mm |

*Continued on next page*

*Continued*

| Reagent type (species) or resource | Designation | Source or reference | Identifiers | Additional information |
|---|---|---|---|---|
| Other | Mono Fiber-Optic Cannula | Doric Lenses, Inc | Cat#: MFC_300/330–0.48_ 3 mm_ZF1.25_FLT | OD: 300 μm, Length: 3 mm |
| Other | Mono Fiber-Optic Cannula | Doric Lenses, Inc | Cat#: MFC_300/330–0.48_ 5 mm_ZF1.25_FLT | OD: 300 μm, Length: 5 mm |
| Other | Mono Fiber-Optic Patch Cable | Doric Lenses, Inc | Cat#: MFP_400/430 /LWMJ-0.48_ 2 m_FC- ZF1.25, Doric Lenses Inc | OD: 400 μm, Length: 2 m |
| Other | Mono Fiber-Optic Patch Cable | Doric Lenses, Inc | Cat#: MFP_300/ 330 /LWMJ-0.48 _1 m_FC-ZF1.25, Doric Lenses Inc | OD: 300 μm, Length: 1 m |

## Experimental animals

Experimental subjects were 8–12 week old 129T2/SvEmsJ::C57Bl/6NTac $Nf1^{+/+}$ and $Nf1^{+/-}$ male and female mice that were generated via the F1 cross of 129T2/SvEmsJ male mice (the Jackson Laboratory Stock No: 002065) and C57Bl/6NTac $Nf1^{+/-}$ female mice (generous gift of Dr. Alcino Silva, UCLA). Animals were group housed (3–4 per group) throughout the duration of the experiment in a vivarium on a 12 hr light/dark cycle (lights off at 0600 hr, lights on at 1800 hr) with *ad libitum* access to food and water. Fluid restricted animals were singly housed, and their water access was limited to 1.5 mL/day. These mice were weighed daily and were returned to *ad libitum* water access if their weight decline was >10% of their pre-restriction weight. Animal husbandry and experimental procedures involving animal subjects were conducted in compliance with the Guide for the Care and Use of Laboratory Animals of the National Institutes of Health and approved by the Institutional Animal Care and Use Committee (IACUC) and by the Office of Laboratory Animal Resources at California Institute of Technology under IACUC protocol 1730. Mice were only excluded from behavioral studies if they could not complete the entire experiment due to health concerns, if there was no dynamic photometry signal 3 weeks after surgery, or if the location of the photometry fiber tip was histologically determined to be outside the LNAc (1 mouse). All behavioral experiments were performed in at least two cohorts to minimize batch effects. All experiments were performed and analyzed blinded to genotype using automated, batched data analysis scripts/software wherever possible to eliminate experimenter bias. Littermate controls were used throughout the study.

## Patch-clamp electrophysiology

Whole-cell patch-clamp recordings were performed as previously described (*Cho et al., 2017*) in acute brain slices. Acute 250 μm-thick horizontal slices that contained the lateral VTA were prepared on a vibratome (VT-1200, Leica Biosystems) from 8 to 12 week $Nf1^{+/+}$ and $Nf1^{+/-}$ mice that had been transcardially perfused with an ice-cold NMDG cutting solution (*Ting et al., 2014*) saturated with 95% $O_2$/5% $CO_2$. Slices were recovered at 32°C in NMDG cutting solution for ten minutes prior to transfer to HEPES recovery artificial cerebrospinal fluid (ACSF) (*Ting et al., 2014*) for an additional 30 min of recovery. During recording, slices were continuously perfused (2.0–3.0 mL/min) with 32°C, 95% $O_2$/5% $CO_2$ -saturated recording ACSF that contained (mM): 125 NaCl, 2.5 KCl, 1.2 $NaH_2PO_4$, 1.2 $MgCl_2$, 2.4 $CaCl_2$, 26 $NaHCO_3$, and 11 glucose. The medial terminal nucleus of the accessory optic track (MT) was used as a visual landmark to delineate the most lateral region of the VTA. Whole-cell patch clamp recordings were obtained using 3–6 MΩ patch pipettes fabricated from borosilicate capillary glass tubing (World Precision Instruments) and backfilled with a potassium gluconate internal solution that contained (mM): 135 K gluconate, 5 KCl, 5 EGTA, 0.5 $CaCl_2$, 10 HEPES, 2 Mg-ATP, and 0.1 GTP. A high-chloride internal solution was used to measure IPSCs and contained (mM): 128 KCl, 20 NaCl, 1 $MgCl_2$ 1 EGTA, 0.3 $CaCl_2$, 10 HEPES, 2 Mg-ATP, and 0.3 GTP. Signals

were amplified and digitized using a MultiClamp 700B amplifier (Molecular Devices, LLC) and Digidata 1440 analog-to-digital converter (Molecular Devices, LLC). Series resistance (Rs) was monitored throughout recording, and data were discarded if the uncompensated Rs exceeded 30 MΩ or the holding current at −70 mV was more negative than −200 pA. Rheobase currents were determined via the injection of a 500 pA ramp current over 500 ms from −60 mV in current clamp mode. $I_h$ currents were measured during seven 2 s, −10 mV hyperpolarizing voltage steps from −60 mV in voltage clamp in the presence of 20 µM bicuculline methiodide and 3 mM kynurenic acid + /- 20 µM forskolin. Spontaneous firing was measured over twenty seconds of gap free recording in I = 0 mode. Spontaneous IPSCs and EPSCs were measured at −70 mV in voltage clamp. IPSCs were recorded in the presence of 3 mM kynurenic acid. Electrophysiological data were sampled at 10 kHz and filtered at 2 kHz with Clampex 10.4 and analyzed in Clampfit 10.7 (Molecular Devices, LLC).

## Surgical procedures

Stereotaxic viral vector injections were performed in mice anesthetized with isoflurane (1–3% in 95% $O_2$/5% $CO_2$ provided via nose cone at 1 L/min) as previously described (Cho et al., 2017). Following anesthesia, preparation and sterilization of the scalp, and exposure of the skull surface, a craniotomy hole was drilled over the LNAc (antero-posterior: 1.2 mm, medio-lateral: 1.6 mm relative to Bregma). 800 nL of the AAV9-hSyn-dLight1.2 vector (titer:~4 × $10^{12}$ viral genomes/mL, produced at the UC Davis Vision Center Vector Design and Packaging Core facility; Addgene # 111068) was delivered into the LNAc (antero-posterior: 1.2 mm, medio-lateral: 1.6 mm, dorso-ventral: −4.2 mm relative to Bregma) using a blunt 33-gauge microinjection needle within a 10 µL microsyringe (NanoFil, World Precision Instruments), a WPI microsyringe pump (UMP3, World Precision Instruments), and pump controller (Micro4, World Precision Instruments) over 10 min. Following viral injection, a 5 mm long, 400 µm outer diameter mono fiber-optic cannula (MFC_400/430–0.48_5 mm_ZF1.25_FLT, Doric Lenses Inc) was lowered to the same stereotaxic coordinates and affixed to the skull surface with C and B Metabond (Parkel Inc) and dental cement. For optogenetic stimulation of the SC during dLight1.2 recordings, mice received a second stereotaxic injection of AAV5-hSyn-ChR2(H134R)-eYFP (UNC Vector Core) in the SC (antero-posterior: −4.0 mm, medio-lateral: 0.5 mm, dorso-ventral: −1.5 mm relative to Bregma), followed by implantation of a 3 mm long, 300 µm mono fiber-optic cannula (MFC_300/330–0.48_3 mm_ZF1.25_FLT, Doric Lenses Inc; antero-posterior: −4.0 mm, medio-lateral: 0.5 mm, dorso-ventral: −1.3 mm relative to Bregma). For optogenetic inhibition of $VTA^{non-Th}$ neurons, 500 nL of a 1:4 mixture of AAV9-Th-PI-Cre-SV40 (gift of James M. Wilson, Addgene viral prep # 107788-AAV9) and AAV-DJ-Ef1α-DO-eNpHR3.0-eYFP-WPRE-pA (gift of Bernardo Sabatini, Addgene # 37087) or AAV-DJ-Ef1α-DO-mCherry-WPRE-pA (gift of Bernardo Sabatini, Addgene # 37119) was injected bilaterally into the VTA (antero-posterior: −3.3 mm, medio-lateral: ± 0.5 mm, dorso-ventral: −4.2 mm relative to Bregma), followed by implantation of 5 mm long, 300 µm mono fiber-optic cannulae (MFC_300/330–0.48_5 mm_ZF1.25_FLT; antero-posterior: −3.3 mm, medio-lateral: ± 1.84 mm, dorso-ventral: −3.59 mm relative to Bregma) at angle of twenty degrees. Mice were given 1 mg/kg buprenorphine SR and 5 mg/kg ketoprofen s.c. intraoperatively and received 30 mg/kg ibuprofen p.o. in their home cage water for five days post-operatively for pain. Mice were allowed a minimum of 14 days for surgical recovery prior to participation in behavioral studies.

## Systemic AAV vector production and administration

In order to create Th-VAST, a PCR fragment containing the 2.5 kb rat tyrosine hydroxylase promoter (Oh et al., 2009) was subcloned into pAAV-ihSyn1-tTA-WPRE (Addgene #99120), replacing the ihSyn promoter through AflII and MluI restriction digest to create pAAV-Th-tTA-WPRE. pAAV-TRE-mRuby-WPRE (Addgene # 99114), pAAV-TRE-mNeonGreen-WPRE, pAAV-TRE-mTurquoise-WPRE (Addgene # 99113), pAAV-Th-GFP-WPRE (Addgene # 99128), pAAV-Ef1α-DO-mCherry-WPRE-pA (Addgene # 37119), and pAAV-Ef1α-DO-NpHR3.0-eYFP-WPRE-pA (Addgene # 37087) constructs were used as previously described (Chan et al., 2017; Saunders et al., 2012). Virus production was performed using a published protocol (Challis et al., 2019). In brief, HEK293T cells were triple transfected using polyethylenimine (PEI) to deliver viral pUCmini-iCAP-PHP.eB (Addgene #103005) or pAAV-DJ-Rep-Cap (VPK-420-DK, Cell Biolabs, Inc), pHelper, and transgene plasmids. Viral particles were harvested from the media and cell pellet and purified over 15%, 25%, 40% and 60% iodixanol (OptiPrep, STEMCELL Technologies, Inc) step gradients. Viruses were concentrated using Amicon

Ultra centrifugal filters (Millipore Sigma), formulated in sterile phosphate buffered saline, and titered with qPCR by measuring the number of DNase I–resistant viral genomes relative to a linearized genome plasmid as a standard. Following viral production and titering, systemic AAV vectors were administered via injection into the retro-orbital sinus during anesthesia with isoflurane (1–3% in 95% $O_2$/5% $CO_2$ provided via nose cone at 1 L/min), followed by administration of 1–2 drops of 0.5% proparacaine to the corneal surface (*Challis et al., 2019*). Note: we have observed some toxicity with doses of AAV-PHP.eB-*Th*-tTA$\geq$1$\times$10$^{11}$ vg/mouse when allowed to express >3 weeks; therefore, we encourage users to perform dosing and time course studies when beginning experiments with *Th*-VAST.

## Fiber photometry

Fiber photometry was used to monitor fluorescent dopamine signals using a custom system as previously described (*Cho et al., 2017*; *Patriarchi et al., 2018*), which allowed for dLight1.2 excitation and emission light to be delivered and collected via the same implanted optical fiber. Our system employed a 490 nm LED (M490F1, Thorlabs, Inc; filtered with FF02-472/30-25, Semrock) for fluorophore excitation and a 405 nm LED for isosbestic excitation (M405F1, Thorlabs Inc; filtered with FF01-400/40-25, Semrock), which were modulated at 211 Hz and 531 Hz, respectively, controlled by a real-time processor (RX8-2, Tucker David Technologies), and delivered to the implanted optical fiber via a 0.48 NA, 400 µm diameter mono fiber optic patch cable (MFP_400/430/LWMJ-0.48_2 m_FC-ZF1.25, Doric Lenses Inc). The emission signal from isosbestic excitation, which has previously been shown to be calcium independent for GCaMP sensors (*Kim et al., 2016*; *Lerner et al., 2015*), was used as a reference signal to account for motion artifacts and photo-bleaching. Emitted light was collected via the patch cable, collimated, filtered (MF525-39, Thorlabs), and detected by a femto-Watt photoreceiver (Model 2151, Newport Co.) after passing through a focusing lens (62–561, Edmunds Optics). Photoreceiver signals were demodulated into dLight1.2 and control (isosbestic) signals, digitized (sampling rate: 382 Hz), and low-pass filtered at 25 Hz using a second-order Butterworth filter with zero-phase distortion. A least-squares linear fit was applied for the 405 nm signal to be aligned with the 490 nm signal. Then, the fitted 405 nm signal was subtracted from 490 nm channel, and then divided by the fitted 405 nm signal to calculate ∆F/F values. The code to perform this function is available at: https://github.com/GradinaruLab/dLight1/blob/master/FP_Session_Processing2.m.

During behavioral experiments, the ∆F/F time-series trace was normalized using a robust z-score ($\frac{signal - signal\ median}{median\ absolute\ deviation}$) to account for data variability across animals and sessions. When fiber photometry was performed during behavioral testing, dLight1.2 signals were synchronized to the beginning of the behavioral session by delivery of TTL pulses (via a TTL pulse generator; OTPG_4, Doric Lenses Inc) to the photometry system.

## Behavioral assays

*Baseline dLight1.2 Measurements*: Mice were tethered to the photometry patch cable, placed in a clean home cage within a sound attenuating box, and allowed to habituate for 5 min (Lafayette Instrument Company). Spontaneous dLight1.2 signals were subsequently recorded for 5 min. This procedure was repeated at least three times per mouse. During pre-treatment experiments, morphine sulfate (5.0 mg/kg s.c.) or saline (s.c.) was administered twenty minutes prior to the onset of each session. Median fluorescence was determined from the processed ∆F/F time-series trace using the median() function in Matlab; peak analysis was performed on the normalized, z-scored trace using a two z-score threshold to minimize contamination by fluorescent noise. Dopamine transients were detected using the findpeaks() function in Matlab, and outputs were averaged within each subject across trials.

*Reward Consumption and Pavlovian Conditioning:* During the twenty-minute sucrose consumption assay, mice were given access to ten 50 µL sucrose (5% w/v) rewards delivered every 60 s (0.5 mL total) via a lick port in a mouse modular test chamber (Model 80015NS, Lafayette Instrument Company) placed within a sound-attenuating box and controlled by ABET II software (Lafayette Instrument Company). Sucrose consumption was characterized by measuring the timing and number of licks at the lick spout, which was measured with an optical lickometer in the lick port. Lick bouts were defined as licking events that exceeded five licks/second and were at least 3 s removed from a

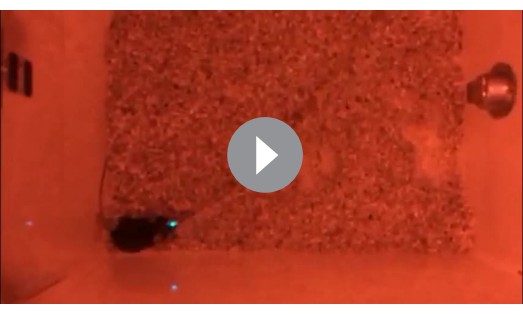

**Video 4.** Behavioral response to optogenetic stimulation of the superior colliculus. https://elifesciences.org/articles/48983#video4

previous lick bout. During the Pavlovian conditioning assay, mice were conditioned to associate the presence of a 10-s conditioned stimulus (CS; illumination of the house-light and delivery of a 60 dB, 3 kHz tone) with delivery of the unconditioned stimulus (US; 50 μL 5% sucrose w/v; 1000 μL total fluid delivery/session) 7 s after the onset of the CS. Each conditioning session consisted of twenty trials with an inter-trial interval randomly drawn from a uniform distribution between 75 and 105 s. On day 11, the sucrose US was omitted to determine dLight1.2 responses to reward omission (Day 11, Trial 1). The number of licks during CS presentation were determined for each trial using ABET II. Because US consumption often occurred during the inter-trial interval, the US timing was determined by identifying the first lick bout after reward presentation during each trial.

*Cued Fear Conditioning:* During the cued fear conditioning task, a ten-second CS (60 dB, 3 kHz tone and house light illumination) immediately preceded a 1 s, 0.4-mA foot shock delivered via the grid floor of the conditioning chamber (Lafayette Instrument Company). The conditioning procedure was completed during 15 consecutive trials with an inter-trial interval randomly drawn from uniform distribution between 75 and 105 s. Mice were videotaped during the assay under dim red light conditions so that the amount of time freezing (defined as the absence of any body movement) and the latency to freeze during each trial could be quantified post hoc by two blinded reviewers.

*Audiovisual Stimulus Exposure:* Mice were placed in a clean home cage in the sound attenuating chamber placed underneath the speaker and house light from the modular conditioning chamber. During each trial, mice were randomly presented with either a ten-second 60 dB, 3 kHz tone or house light illumination, which had equal probability of selection. Each session consisted of twenty trials, and the inter-trial interval was randomly drawn from a uniform distribution between 75 and 105 s.

*Social Interaction and Social Preference Assay:* Test mice were placed in a clean cage and allowed to freely interact with a juvenile, sex-matched, novel, conspecific probe mouse. Social interactions were videotaped, and the onset of recording was synchronized with the photometry signal via delivery of TTL pulses. The onset of each social interaction was defined as the initiation of physical contact between mice and was terminated when mice physically disengaged. The social preference assay was performed in a 50 cm x 50 cm square, white acrylic arena that contained two identical, wire mesh-enclosed social interaction vestibules placed in the center of opposite walls of the arena. During each thirty-minute testing session, a juvenile, sex-matched, novel, conspecific probe mouse was placed in one of the vestibules, and the position of the test mouse was tracked with an overhead camera using EthoVision XT 10 (The Noldus Company). The location of the probe mouse was alternated between trials.

*Looming Stimulus Assay:* The looming stimulus assay was performed as previously described (*Yilmaz and Meister, 2013*). Mice were acclimated in an 87 cm x 47.5 cm x 30 cm (h) infrared-transmitting black acrylic arena in the presence of a nest/shelter for at least five minutes. The overhead looming stimulus was presented when the animal was in the center of the arena. The looming stimulus covered 5 degrees of the animal's visual field initially, expanded up to 50 degrees, and was presented five times separated by 1 s pauses on a gray background. For optogenetic

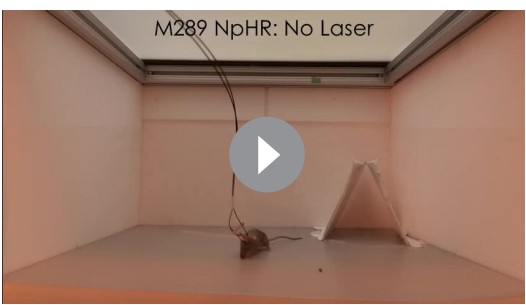

**Video 5.** Behavioral response to looming stimulus in *Nf1*[+/-] mouse with or without optogenetic inhibition of VTA[non-*Th*] neurons. https://elifesciences.org/articles/48983#video5

manipulations, the looming stimulus was presented as described above in an arena that was optimized for behavioral testing in tethered mice that measured 50 cm x 35 cm x 30 cm (h). In order to accommodate optical patch cables, the arena featured a rectangular opening adjacent to the overhead screen and a triangular nest/shelter with a 2 cm slit at the apex. During testing, laser pulses (532 nm, 5 mW, 20 Hz, 5 ms pulse width) were delivered via the implanted optical fibers, which were coupled to optical patch cables (MFP_300/330/LWMJ0.48_1 m_FC-ZF1.25, Doric Lenses Inc) connected to a 532 nm laser (Changchun New Industries [CNI] Model with PSU-H-LED). Stimulation began 1.5 s prior to looming onset and continued throughout the stimulus presentation. Mice were videotaped throughout so that time to react to the disc, escape location, latency to the first freezing episode, duration of the first freezing episode, and amount of time spent freezing during the first minute after exposure could be quantified by a blinded reviewer.

*Optogenetic Stimulation of the Superior Colliculus*: Mono fiber-optic patch cables were connected to implanted optical fibers for fiber photometry in the LNAc and optogenetic stimulation of the ipsilateral vSC. The vSC patch cable was connected to a 473 nm laser (Changchun New Industries [CNI] Model with PSU-H-LED) and controlled by a Doric Lenses OTPG_4 pulse generator that was triggered by the ABETII software. Mice were allowed to habituate in a clean home cage within the Lafayette sound-attenuating chamber for 5 min prior to the start of the experiment. Following habituation, dLight1.2 signals were recorded during twenty trials in which the vSC was stimulated with 2 s of 20 Hz, 5 ms, 473 nm laser pulses. The inter-trial interval was randomly drawn from a uniform distribution between 75 and 105 s.

## Histology

Free-floating brain sections were blocked for 1 hr in 10% normal donkey serum (Millipore-Sigma, S30-M) in phosphate buffered saline (PBS) with 0.1% Triton X-100 at room temperature, then incubated with primary antibody diluted in the blocking buffer overnight at 4°C. Sections were washed three times for 15 min in PBS. Secondary antibodies were diluted in blocking buffer, and brain sections were incubated in the secondary antibody solution for 2 hr at room temperature. Sections received three 15 min washes in PBS prior to mounting on glass slides with ProLong Diamond antifade mounting medium (ThermoFisher Scientific, P36965) or RIMS (refractive index matching solution; RI = 1.46) (*Yang et al., 2014*). VAST sections were optically cleared overnight in RIMS prior to mounting. The following antibodies/dilutions were used: rabbit anti-tyrosine hydroxylase (EMD Millipore, AB152, 1:1000; for VAST), monocloncal mouse anti-tyrosine hydroxylase (ImmunoStar, 22941, 1:1000; for TH quantification, somata tracing, and cell counting), polyclonal chicken anti-GFP (Aves, GFP-1020, 1:1000), Alexa Fluor 488-conjugated donkey anti-chicken IgY F(ab')two fragment (Jackson ImmunoResearch, 703-546-155, 1:1000), Alexa Fluor 647-conjugated donkey anti-rabbit IgG Fab fragment (Jackson ImmunoResearch, 711-606-152, 1:1000), and Alexa Fluor 647-conjugated donkey anti-mouse IgG Fab fragment (Jackson ImmunoResearch, 711-607-003, 1:1000). Histological images were obtained using either a Keyence BZ-X fluorescence microscope or Zeiss 880 confocal microscope. Images were analyzed using BZ-X Analyzer software (Keyence Corporation), ImageJ, and/or Imaris (Bitplane). VTA and SNc cell counting was performed on 100 mm, coronal histological sections from Bregma $-3.2$ to $-3.6$ μm (AP) and averaged across sections within each mouse.

## Statistical analysis

Statistical analysis was performed using Matlab (MathWorks, Inc) and GraphPad Prism 7 (GraphPad Software, Inc). All statistical tests performed on data presented in the manuscript are stated in the figure captions and provided in detail in *Source data 1*. For each experiment, statistical tests were chosen based on the structure of the experiment and data set. No outliers were removed during statistical analysis. Sample sizes estimates were based on published behavioral and electrophysiological literature that utilized the 129T2/SvEmsJ::C57Bl/6NTac $Nf1^{+/-}$ mouse model; this was within a range commonly employed by researchers in our field using similar techniques and that which was determined via the sampsizepwr() function in Matlab. Parametric tests were used throughout the manuscript; for sIPSC and ESPC data, non-parametric Mann Whitney U tests were also reported because sIPSC frequency was non-normally distributed for both genotypes (as determined by the D'Agostino and Pearson normality test). In this case, the results of parametric and non-parametric hypothesis tests were congruent (see *Figure 3* caption). When analysis of variance (ANOVA; one-way, two-way,

and/or repeated measures) was performed, multiple comparisons were corrected using the Bonferroni correction. Multiple t-tests were corrected with the two-stage linear step-up procedure of Benjamini, Krieger and Yekutieli with a false discovery rate of 5%.

## Data and materials availability

Viral vector plasmids used in this study are available on Addgene at http://www.addgene.org/Viviana_Gradinaru/. Codes used for fiber photometry signal extraction and analysis are available at https://github.com/GradinaruLab/dLight1. Source data is available at www.doi.org/10.7303/syn18904024.

## Acknowledgements

We acknowledge Dr. Ginger Milne and the Vanderbilt Neurochemistry Core (Vanderbilt University School of Medicine) for HPLC analysis; Dr. Markus Meister for expertise in conducting the looming stimulus assay; Dr. Daniel Wagenaar and the Caltech Neurotechnology Center (California Institute of Technology) for technical assistance; George R Hudson for assistance with *Th*-VAST experiments; Varun Wadia and Jaeyoung Kang for assistance with fear conditioning and social preference assays; Aditya Nair for image analysis; and the Beckman Institute for CLARITY, Optogenetics and Vector Engineering Research (CLOVER, California Institute of Technology, clover.caltech.edu) for assistance with vector design and production. This work is funded by NIH Director's New Innovator Award IDP20D017782-01, NIH Presidential Early Career Award for Scientists and Engineers (PECASE), NIH BRAIN RF1MH117069, NSF NeuroNex Technology Hub 1707316, the Heritage Medical Research Institute, and the Tianqiao and Chrissy Chen Institute for Neuroscience (VG); NIH Awards U01NS103522 and DP2MH107056 (LT); Children's Tumor Foundation Young Investigator Award 2016-01-006 (JER); and a PGS-D from the National Science and Engineering Research Council (NSERC) of Canada (GMC).

## Additional information

### Funding

| Funder | Grant reference number | Author |
| --- | --- | --- |
| National Institutes of Health | IDP20D017782-01 | Viviana Gradinaru |
| National Institutes of Health | PECASE | Viviana Gradinaru |
| National Institutes of Health | RF1MH117069 | Viviana Gradinaru |
| National Science Foundation | 1707316 | Viviana Gradinaru |
| Heritage Medical Research Institute | | Viviana Gradinaru |
| Tianqiao and Chrissy Chen Institute for Neuroscience | | Viviana Gradinaru |
| National Institutes of Health | U01NS103522 | Lin Tian |
| National Institutes of Health | DP2MH107056 | Lin Tian |
| Children's Tumor Foundation | Young Investigator Award 2016-01-00 | J Elliott Robinson |
| Natural Sciences and Engineering Research Council of Canada | Postgraduate Scholarship-Doctoral | Gerard M Coughlin |

The funders had no role in study design, data collection and interpretation, or the decision to submit the work for publication.

### Author contributions

J Elliott Robinson, Conceptualization, Data curation, Software, Formal analysis, Investigation, Visualization, Methodology, Writing—original draft, Project administration, Writing—review and editing;

Gerard M Coughlin, Investigation, Methodology, Writing—review and editing; Acacia M Hori, Formal analysis, Investigation; Jounhong Ryan Cho, Software, Methodology, Writing—review and editing; Elisha D Mackey, Resources, Methodology; Zeynep Turan, Investigation, Methodology; Tommaso Patriarchi, Lin Tian, Methodology, Writing—review and editing; Viviana Gradinaru, Conceptualization, Supervision, Funding acquisition, Project administration, Writing—review and editing

**Author ORCIDs**
J Elliott Robinson ⓘ https://orcid.org/0000-0001-9417-3938
Lin Tian ⓘ http://orcid.org/0000-0001-7012-6926
Viviana Gradinaru ⓘ https://orcid.org/0000-0001-5868-348X

**Ethics**
Animal experimentation: Animal husbandry and experimental procedures involving animal subjects were conducted in compliance with the Guide for the Care and Use of Laboratory Animals of the National Institutes of Health and approved by the Institutional Animal Care and Use Committee (IACUC) and by the Office of Laboratory Animal Resources at the California Institute of Technology under IACUC protocol 1730.

**Decision letter and Author response**
Decision letter https://doi.org/10.7554/eLife.48983.sa1
Author response https://doi.org/10.7554/eLife.48983.sa2

# Additional files

**Supplementary files**
• Source data 1. Summary of statistical analysis.

• Supplementary file 1. Trial-by-trial correlations between peak dLight1.2 responses to CS presentation and behavioral measures.

• Supplementary file 2. Correlations between dLight1.2 responses and behavioral measures.

• Transparent reporting form

**Data availability**
Viral vector plasmids used in this study are available on Addgene at http://www.addgene.org/Viviana_Gradinaru/. Codes used for fiber photometry signal extraction and analysis are available at https://github.com/GradinaruLab/dLight1. Source data is available at https://doi.org/10.7303/syn18904024.

The following dataset was generated:

| Author(s) | Year | Dataset title | Dataset URL | Database and Identifier |
|---|---|---|---|---|
| Robinson JE, Coughlin GM, Hori AM, Cho JR, Mackey ED, Turan Z, Patriarchi T, Tian L, Gradinaru V | 2019 | Source Data: Optical dopamine monitoring with dLight1 reveals mesolimbic phenotypes in a mouse model of neurofibromatosis type 1 | https://doi.org/10.7303/syn18904024 | Synapse, 10.7303/syn18904024 |

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
