## [Decision Letter]

Thank you for submitting your article "Optical dopamine monitoring with dLight1 reveals mesolimbic phenotypes in a mouse model of neurofibromatosis type 1" for consideration by *eLife*. Your article has been reviewed by two peer reviewers, and the evaluation has been overseen by a Reviewing Editor and Eve Marder as the Senior Editor. The following individual involved in review of your submission has agreed to reveal their identity: Camilla Bellone (Reviewer #1).

The reviewers have discussed the reviews with one another and the Reviewing Editor has drafted this decision to help you prepare a revised submission.

Summary:

In this manuscript the authors, using the genetically encoded optical dopamine sensor dLight1, characterized the mesolimbic DA circuitry in Neurofibromatosis (NF1). Monitoring dopamine dynamics in the lateral NAc the authors have observed decreased spontaneous DA transient activity in *Nf1*^+/-^ mice compared to control. Changes in electrophysiological properties were accompanied by differences in soma size but no differences in dendritic complexity or total neurite length. By investigating excitation/inhibition balance, the authors reported an increase in sIPSC frequency. Since spontaneous firing was rescued by PTX, the authors concluded that E/I imbalance is the mechanism gating *Nf1*^+/-^ DA excitability ex vivo that may contribute to reduced baseline DA transient in NAc. Finally, by monitoring DA dynamics during fear conditioning, the authors have demonstrated that the pattern of DA transmission in response to US and CS is different between genotypes. These deficits depend on the visual stimulus. Indeed *Nf1*^+/-^ mice exhibit larger responses to light compared to *Nf1*^+/+^ mice. To test whether these changes reflect an increased motivational salience of an alerting stimulus, the authors performed a looming task. *Nf1*^+/-^ mice were more likely to escape to the shelter at stimulus onset and exhibited shorter latency to the first freezing episode.

Overall this is an excellent study, well presented and well designed. The findings are original and novel. A few points need to be addressed to strengthen some of the results.

Essential revisions:

1) When the authors monitor activity and electrophysiological properties of DA neurons in the VTA, they do not account for output specificity. Are the changes in DA transient in the NAc specific for the lateral part?

2) The change in the E/I balance is interesting but the findings are not integrated in the context of the behavior. Is this E/I imbalance the cause of hypersensitivity to visual stimuli? Is the E/I imbalance a consequence of increased activity of GABA interneurons within the VTA? In this case, the authors could act on GABA neurons and observe whether response to visual stimuli is rescued.

3) The finding that *Nf1*^+/-^ mice are more sensitive to salient visual stimuli is very interesting. Why would this deficit be relevant only in response to aversive stimuli? What would happen if the authors would use a visual cue for reward conditioning instead of a tone?

4) The manuscript presents both in-vitro and in-vivo results in relation to the possible effects of the deletion of a single *Nf1* gene on dopaminergic transmission. However, the connections between the two parts of the study – i.e. the behavioral and the in-vitro measurements remain unsubstantiated. The authors find results which are indicative of stronger GABAergic input to VTA dopaminergic neurons; they also refer to published work showing that subthreshold doses of picrotoxin in-vivo improve the cognitive performance of NF1 model mice. It appears to me that connecting these two observations would significantly increase the impact of this manuscript. For example, the authors can determine whether administration of subthreshold picrotoxin rescues the in-vivo alterations in dopaminergic transmission as measured using dLight.

5) Could the authors explain why dLight measurements in Figure 1E produce negative values? In this respect, I would expect the authors to explain in a more complete manner the calculation of the dLight values that they report. The authors indicate that they use the isosbestic point of dLight, but the details of how dLight values are calculated are not spelled out for the reader to appreciate.

---

## [Author Response]

Essential revisions:1) When the authors monitor activity and electrophysiological properties of DA neurons in the VTA, they do not account for output specificity. Are the changes in DA transient in the NAc specific for the lateral part?

This is an excellent point, and we apologize for not making it clear in the Materials and methods section. All dopaminergic neurons recorded in ventral tegmental area were recorded in the lateral ventral tegmental area, near the medial terminal nucleus of the accessory optic tract (MT) in horizontal brain slices. We targeted this region because the work of Lammel and others has shown that the LNAc preferentially receives projections from the lateral VTA. We have now specified these changes in the manuscript:

“In order to further parse differences in spontaneous dopaminergic transient activity, we performed whole-cell patch clamp electrophysiological recordings in acute midbrain slices that contained the lateral ventral tegmental area (Figure 1F), which is the main source of dopaminergic projections to the LNAc (Lammel et al., 2011)”.

In the Materials and methods, we’ve added the following sentence:

“The medial terminal nucleus of the accessory optic track (MT) was used as a visual landmark to delineate the most lateral region of the VTA.”

2) The change in the E/I balance is interesting but the findings are not integrated in the context of the behavior. Is this E/I imbalance the cause of hypersensitivity to visual stimuli? Is the E/I imbalance a consequence of increased activity of GABA interneurons within the VTA? In this case, the authors could act on GABA neurons and observe whether response to visual stimuli is rescued.

We agree with the reviewers that these experiments would provide an important bridge between ex vivo electrophysiology and behavioral phenotypes in *Nf1*^+/-^ mice. To address this concern, we have used an intersectional strategy to target non-dopaminergic neurons in the ventral tegmental area for optogenetic rescue in *Nf1*^+/-^ mice. In the VTA, ~60-65% of neurons are dopaminergic, while ~30-35% are GABAergic, and 2-3% are glutamatergic (Morales and Root, 2014; Nair-Roberts et al., 2008; Pignatelli and Bonci, 2015). In order to target non-dopaminergic cells in the VTA, we co-delivered AAV9-*rTh*-PI-Cre and AAV-DJ-*Syn*-DO-eNpHR3.0-eYFP (*Nf1*^+/-^ mice) or AAV-DJ-*Syn*-DO-mCherry-eYFP (*Nf1*^+/+^ mice) into the VTA, followed by implantation of bilateral 300-μm optical fibers. In this scenario, Cre recombinase is expressed in *Th*-positive cells, which leads to inactivation of the open reading frame of the double-floxed transgene in dopaminergic neurons, while the correct transgene orientation is preserved in non-*Th*-expressing, primarily GABAergic, neurons (Figure 6H; Figure 6—figure supplement 3). This strategy was chosen because of the lack of robust cell type-specific promoters targeting GABAergic neurons in the ventral midbrain and our inability to cross knockout mice to a *Vgat*-Cre line (due to phenotype dependence on the hybrid genetic background).

In the absence of photoinhibition, fiberoptic-tethered VTA*^Th^*^-Off-NpHR-eYFP^*Nf1*^+/-^ mice were more likely to escape to the available shelter and freeze with short latency following looming stimulus presentation compared to VTA*^Th^*^-Off-mCherry^*Nf1*^+/+^ mice (Figure 6I), which replicates our previous behavioral findings in non-tethered mice (Figure 6F). Delivery of 532-nm light (5 mW, 30Hz, 20-ms pulse width) to the VTA during looming stimulus presentation rescued these phenotypes to wildtype levels in VTA*^Th^*^-Off-NpHR-eYFP^*Nf1*^+/-^ mice without affecting the performance of VTA*^Th^*^-Off-mCherry^*Nf1*^+/+^ mice. Overall, this experiment suggests that optogenetic inhibition of VTA*^non-Th^* neurons, >90% of which are GABAergic, is sufficient to reverse salient visual stimulus responses in *Nf1*^+/-^ mice. Additionally, these data are consistent with recent findings that VTA GABA neurons modulate defensive responses to a looming stimulus in mice (Zhou et al., 2019).

In the revised manuscript, we have added a new section describing these results (“Optogenetic inhibition of non-dopaminergic neurons in the VTA of *Nf1*^+/-^ mice during looming stimulus presentation”), two new panels in Figure 6 (H-I), additional confocal images in Figure 6—figure supplement 3, and a new video (Video 5). The corresponding methodological information can be found in the Materials and methods.

3) The finding that Nf1^+/-^ mice are more sensitive to salient visual stimuli is very interesting. Why would this deficit be relevant only in response to aversive stimuli? What would happen if the authors would use a visual cue for reward conditioning instead of a tone?

As detailed in Figure 4, reward conditioning was performed with the same visual cue used during cued fear conditioning. Although increased dopaminergic responses to the CS were seen in both naïve and conditioned *Nf1^+/-^* mice, task performance did not differ between phenotypes. We believe the reason for this is two-fold: first, as was shown during the cued fear conditioning/interleaved light experiments, *Nf1^+/-^* mice are fully capable of forming cue-outcome associations, so any deficits observed in the Pavlovian conditioning assay induced by overhead light exposure would be predicted to be caused by an escape or exploratory response that overrode reward seeking. Because chronically water-restricted mice are highly motivated to consume liquid rewards, this drive was likely too strong to be perturbed by light delivery alone. We agree that repeating reward conditioning with a light-only CS would be an interesting experiment, although this is not possible with our current setup, as our operant conditioning chamber makes a reward-predictive auditory click during sucrose delivery that served as a tone-only CS during conditioning experiments. As we showed in Figure 4—figure supplement 2C-E, mice develop a conditioned dopaminergic response to this pump cue that becomes highly correlated with learning. Future studies will be necessary to further parse the nature of visual processing deficits in *Nf1^+/-^* mice and determine how these cues modulate reward seeking behaviors; at this time, we feel (and hope the reviewer will agree) additional efforts on this topic are beyond the scope of the current study.

4) The manuscript presents both in-vitro and in-vivo results in relation to the possible effects of the deletion of a single Nf1 gene on dopaminergic transmission. However, the connections between the two parts of the study – i.e. the behavioral and the in-vitro measurements remain unsubstantiated. The authors find results which are indicative of stronger GABAergic input to VTA dopaminergic neurons; they also refer to published work showing that subthreshold doses of picrotoxin in-vivo improve the cognitive performance of NF1 model mice. It appears to me that connecting these two observations would significantly increase the impact of this manuscript. For example, the authors can determine whether administration of subthreshold picrotoxin rescues the in-vivo alterations in dopaminergic transmission as measured using dLight.

As mentioned above, we agree with the reviewers that connecting ex vivo results to in vivo dopaminergic measurements with dLight1 would greatly improve the interpretability of our data. As such, we have measured spontaneous dLight1.2 transient activity in the LNAc with fiber photometry in a new cohort of *Nf1^+/-^* mice following pharmacological manipulation with either subthreshold doses of picrotoxin (0.01 mg/kg, i.p.) or a μ-opioid receptor (μOR) agonist (morphine sulfate, 5 mg/kg, s.c.). A μOR agonist was chosen because recent data from Lüscher and colleagues (Corre et al., 2018) showed that intravenous opioids increase NAc dLight1.1 signals via disinhibition of VTA dopaminergic neurons by directly inhibiting local GABAergic neurons. Similarly, we found that administration of morphine significantly enhanced dLight1.2 transient rate (0.54 ± 0.03 Hz) but not event magnitude or width in *Nf1^+/-^* mice relative saline controls to levels that were not significantly different to baseline data from *Nf1^+/+^* mice (0.50 ± 0.02) These data are described in the subsection “*Nf1^+/-^*putative dopaminergic neurons exhibit excitation/inhibition imbalance in the ventral tegmental area”, and displayed in Figure 3—figure supplement 2. Additionally, raw baseline data have been incorporated into Figure 1E.

We also tested the effects of either acute or chronic subthreshold (0.01 mg/kg, i.p.) picrotoxin on spontaneous dLight1.2 activity according to the dosing schedule in (Cui et al., 2008). However, we did not find a statistical difference versus saline control for all spontaneous dopamine transient measures. Review of the literature suggests that this dose may be too low to have robust effects on its own [e.g. (Fernandez et al., 2007; Koechling et al., 1991; Ramwell and Shaw, 1965)], as many studies required doses of at least 0.1 – 0.3 mg/kg i.p. in order to modulate behavior. Unfortunately, we were unable to test higher doses of systemically administered picrotoxin, a known convulsant, due to health concerns by the Caltech Institutional Animal Care and Use Committee – seeking approval involved delays that were beyond the revision timelines.

5) Could the authors explain why dLight measurements in Figure 1E produce negative values? In this respect, I would expect the authors to explain in a more complete manner the calculation of the dLight values that they report. The authors indicate that they use the isosbestic point of dLight, but the details of how dLight values are calculated are not spelled out for the reader to appreciate.

Upon reviewing our Materials and methods section in the original submission, we realized that we have accidentally omitted important details regarding the calculation of the ΔF/F trace. We apologize for this error and understand the reviewer’s concern. We have corrected this in the revised manuscript, which now reads:

“A least-squares linear fit was applied for the 405-nm signal to be aligned with the 490-nm signal. Then, the fitted 405-nm signal was subtracted from 490-nm channel, and then divided by the fitted 405-nm signal to calculate ΔF/F values.”

This method of processing fiber photometry data is widely employed [e.g. (Augustine et al., 2018; Cho et al., 2017; Lerner et al., 2015; Muir et al., 2018)] and is used to minimize motion artifacts and the effect of photo-bleaching, which we have noted in the manuscript. Because of its broad use in the community, this data normalization strategy has been integrated into commercial systems sold by Tucker-Davis Technologies and Doric Lenses.

In order to make the manuscript as transparent as possible, we have edited the text to provide a direct link to the signal processing Matlab code in the Fiber Photometry section of the Materials and methods: https://github.com/GradinaruLab/dLight1/blob/master/FP_Session_Processing2.m

Regarding negative ΔF/F, this occurs when the fitted 405-nm signal is subtracted from the fitted 490-nm signal. As you can see in the example trace that has been included in the new Figure 1—figure supplement 1, the signal-independent product of excitation at 405-nm establishes a baseline around which dynamic, dopamine-dependent 490-nm signal oscillates. Thus, when the ΔF calculation is performed, negative values are common.

For our study, we chose to apply a robust z-score to the ΔF/F data to minimize session-to-session variability in fluorescence in longitudinal behavioral studies and eliminate negative baseline values. This method was used previously in original dLight paper when conducting photometry over many days of Pavlovian conditioning [see Figure 4, (Patriarchi et al., 2018)]. As shown in Figure 6—figure supplement 1, this did not affect the underlying trace structure or change the main conclusion in the manuscript (i.e. that *Nf1*^+/-^ mice have more robust dopaminergic responses to salient stimuli).